# ON THE LIMITATION AND REDUNDANCY OF TRANSFORMERS: A RANK PERSPECTIVE

## ABSTRACT

Transformers have showcased superior performance across a variety of real-world applications, particularly leading to unparalleled successes of large foundation models. However, the overall computation and memory loads of these large models trained on web-scale datasets are considerably increasing, calling for more *efficient* learning methods. In this work, we step towards this direction by exploring the architectural limitation and *redundancy* of Transformers via investigating the ranks of attention score matrices. On one hand, extensive experiments are conducted on various model configurations (model dimensions, heads, layers, etc) and data distributions (both synthetic and real-world datasets with varied sequence lengths), uncovering two key properties: The attention rank is eventually upper bounded (limitation) and gets saturated (redundancy), as the head dimension $d_h$ increases. We call them the *low-rank barrier* and *model-reduction effect*, respectively. Most importantly, the redundancy appears that *both the attention rank and learning performance simultaneously get marginal enhancements when increasing modeling parameters*. On the other hand, we provide rigorous demonstrations for these observations under idealized settings through a fine-grained mathematical analysis, highlighting (i) a consistent theoretical upper bound ($\approx 0.63n$, $n$: the sequence length) on the attention rank (regardless of $d_h$) given random weights; (ii) a critical position of the rank saturation ($d_h = \Omega(\log n)$). These results contribute to the principled understanding and assessment of Transformers' model capacity and efficiency, and are also successfully verified in practical applications such as multi-head *latent* attention (MLA) applied in DeepSeek-V3.

## 1 INTRODUCTION

In recent years, Transformer-based neural network models have reshaped the landscape of machine learning, demonstrating unparalleled successes across a myriad of applications including natural language processing (NLP) (Vaswani et al., 2017; Devlin et al., 2019; Raffel et al., 2020; Radford et al., 2018; Rae et al., 2021; Dehghani et al., 2023; Touvron et al., 2023; Liu et al., 2019; Hao et al., 2020; Liu et al., 2021; Yuan et al., 2022), computer vision (CV) (Chen et al., 2021b; Wang et al., 2022; Liang et al., 2021; Lu et al., 2022; Zhu et al., 2021; Wang et al., 2021), audios (Sung et al., 2022; Tsimpoukelli et al., 2021; Li et al., 2022), interdisciplinary sciences (Jumper et al., 2021), and so on. The core architecture module, anchored by the so-called attention mechanism, has been proved as a cornerstone particularly in capturing sequential relationships with intricacies and nuances.

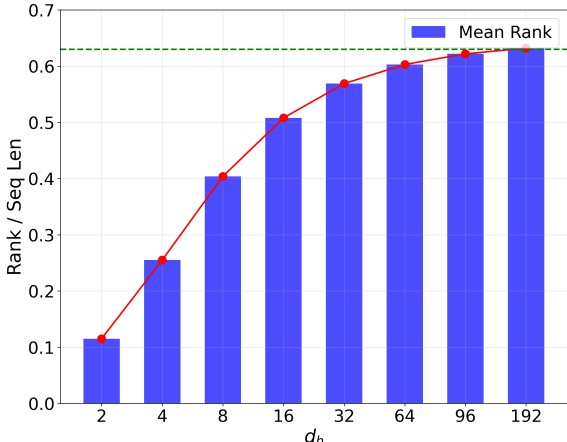

Figure 1: A typical phenomenon of the attention rank of an initialized Transformer for different head dimensions $d_h$. One can observe that the attention rank gets saturated when increasing head dimensions. More importantly, this pattern of diminishing returns also consistently appears for the learning performance, where the test accuracy simultaneously gets marginal enhancements when increasing head dimensions (see Figure 3a and 3b).

Mathematically, the central attention mechanism is designed to weigh the significance and correlations of input sequences via, e.g. inner products between trainable transformations on inputs (e.g. tokens), which is formulated as the attention score matrices. As a fundamental algebra concept, the matrix rank is supposed to impact the capacity (expressive ability) and learning performance of the attention mechanism and hence Transformer models. Particularly, an important phenomenon called the *low-rank bottleneck* is uncovered by numerous recent works (Kanai et al., 2018; Bhojanapalli et al., 2020; Dong et al., 2021; Lin et al., 2022), and several Transformer-based variants aim to reduce the computational and memory overheads of modeling long sequences from the perspective of attention ranks (Chen et al., 2021a; Wang et al., 2020; Hu et al., 2022; Guo et al., 2019; Lin et al., 2022). However, these studies in general (i) are insufficient to quantitatively characterize the attention rank's *limitation* (i.e. deriving low-rank upper bounds); (ii) lack theoretical analysis of the attention rank's *redundancy* (i.e. model-reduction effect). Based on (i), (ii) is straightforwardly applicable in practice, particularly in the current era of large foundation models, where the pre-training efficiency on notably large models on web-scale datasets turns out a remarkable problem.

In this work, we make an initial step towards this direction by studying the limitation and redundancy of general Transformers from the perspective of attention ranks. Figure 1 shows a typical experimental observation in the present work, focusing on the variation of attention ranks with respect to the pivotal head dimension ($d_h$). We observe that: (i) The attention rank increases with the head dimension. As $d_h$ increases within relatively small values, the increment of attention ranks is significant; (ii) For appropriately large values of $d_h$, further increases in $d_h$ lead to a *diminishing return* in the enhancement of attention ranks, with an ultimate upper bound of approximately $0.63n$, which is away from the full rank $n$ ($n$: sequence length and attention matrix size).

Extensive experiments are performed, which consistently demonstrate these observations across various model and data settings, including varied model dimensions, different heads and layers, a variety of data distributions with increasing sequence lengths for both synthetic and real-world datasets. Theoretically, a fine-grained mathematical analysis is provided to rigorously support these experimental observations in a quantitative manner, including that (i) the attention rank has a consistent theoretical upper bound ($\approx 0.63n$) for any $d_h$, which shows the existence of the low-rank barrier ($n$ is the full-rank); (ii) when $d_h = \Omega(\log n)$, the attention rank gets saturated in the sense that further increasing the head dimension leads to diminishing rank enhancement. This study focuses on the model biases inherently in Transformer models, and the developed results not only shed light on the internal dynamics of Transformers, but also provide new insights to evaluate the model capacity and efficiency.

Our main contributions are summarized as follows:

1. Empirically, under extensive settings for general Transformer models and real-world datasets, it is shown that as the head dimension $d_h$ increases, the attention rank rises as expected, but the increment slows down significantly and eventually gets saturated, without reaching the full-rank (for appropriately large $d_h$). More importantly, both the attention rank and learning performance simultaneously get marginal enhancements when increasing modeling parameters, implying principled model redundancy.

2. Theoretically, given random weights, mathematical estimates are established on the barrier of attention ranks, with an upper bound of approximately $0.63n$ (aligned with experimental observations). Moreover, after the critical position $d_h = \Omega(\log n)$ (also numerically verified), the attention rank gets saturated with negligible increments even by significantly increasing head dimensions.

The rest of this paper is organized as follows. Section 2 provides fundamental observations with various experiments and ablation studies. Section 3 includes the fine-grained mathematical analysis on the attention rank. Section 4 further verifies the developed results on real-world datasets. In Section 5, we discuss the related work centering around the attention rank. All the details of proofs and supplementary experiments can be found in the appendix.

**Notations** Throughout this paper, we use normal letters to denote scalars. Boldfaced lower-case/capital letters are reserved for vectors/matrices. Let $[n] := \{1, 2, \ldots, n\}$ for $n \in \mathbb{N}_+$. Let $\|\mathbf{x}\|_p := \left(\sum_{i=1}^n |x_i|^p\right)^{1/p}$ be the $\ell^p$-norm for $\mathbf{x} \in \mathbb{R}^n$ and $p \in [1, \infty]$, and $\|\mathbf{A}\|_F := \left(\sum_{i=1}^m \sum_{j=1}^n a_{ij}^2\right)^{1/2}$ be the Frobenius norm for $\mathbf{A} \in \mathbb{R}^{m \times n}$. Denote the standard basis of $\mathbb{R}^n$ by $\{\mathbf{e}_i\}_{i=1}^n$, i.e., $\mathbf{e}_i$ is the vector of all zeros except that the $i$-th position is 1. Let $\mathbf{0}_n \in \mathbb{R}^n$ be the vector of all zeros. For a probability space $(\Omega, \mathcal{F}, \mathbb{P})$, the probability of a measurable event $E \in \mathcal{F}$ is $\mathbb{P}(E)$. Let $\mathcal{N}(\boldsymbol{\mu}, \boldsymbol{\Sigma})$ be the multivariate normal distribution defined on $\mathbb{R}^n$, where $\boldsymbol{\mu} \in \mathbb{R}^n$ is the expectation and $\boldsymbol{\Sigma} \in \mathbb{R}^{n \times n}$ is the covariance. We use the big-O/big-Omega notation $f(n) = O(g(n))/f(n) = \Omega(g(n))$ to represent that $f$ is bounded above/below by $g$ asymptotically, i.e., there exists $c > 0, n_0 \in \mathbb{N}_+$ such that $f(n) \leq cg(n)/f(n) \geq cg(n)$ for any $n \geq n_0$.

For Transformers, let $\mathbf{X} = [\mathbf{x}_1, \mathbf{x}_2, \ldots, \mathbf{x}_n]^\top \in \mathbb{R}^{n \times d}$ denote the input sequence with the length $n$ and dimension $d$. We use $h$ to represent the number of attention heads and $d_h$ as the head dimension (typically, $d = h \times d_h$). For head $i \in [h]$, let $\mathbf{K}^{(i)}, \mathbf{Q}^{(i)}, \mathbf{V}^{(i)} \in \mathbb{R}^{n \times d_h}$ be the key, query, and value matrices, and $\mathbf{W}_k^{(i)}, \mathbf{W}_q^{(i)}, \mathbf{W}_v^{(i)} \in \mathbb{R}^{d \times d_h}$ are the corresponding weight matrices. When focusing on a single head, we drop the superscripts and define the key-query pair as $(\mathbf{K}, \mathbf{Q}) = (\mathbf{X}\mathbf{W}_k, \mathbf{X}\mathbf{W}_q)$ with trainable parameters $\boldsymbol{\theta} := (\mathbf{W}_k, \mathbf{W}_q) \in \mathbb{R}^{d \times d_h} \times \mathbb{R}^{d \times d_h}$, where the rows are $\mathbf{k}_i^\top = \mathbf{x}_i^\top \mathbf{W}_k$ and $\mathbf{q}_i^\top = \mathbf{x}_i^\top \mathbf{W}_q$ for $i = 1, 2, \ldots, n$. The attention matrix is $\mathbf{Attn}(\mathbf{X}; \boldsymbol{\theta}) := \text{softmax}\left(\mathbf{Q}\mathbf{K}^\top / T\right) \in \mathbb{R}^{n \times n}$ with the temperature $T > 0$.

## 2 MOTIVATING SIMULATIONS

In this section, we provide detailed experiments on general Transformers in various settings to examine the rank of attention matrices. To facilitate comparisons and analysis, we report the ratio of attention ranks over sequence lengths (rank/seq len) rather than the absolute rank values to eliminate the interference caused by varied sizes of attention matrices across different sequence lengths.

### 2.1 BASIC PHENOMENA

First, we test general Transformer models to examine the variations of their attention ranks given various head dimensions.

**Setup.** We use a standard one-layer Transformer encoder block with $d_{\text{model}} = d = 384$ and a feed-forward hidden dimension of 512, and select the head dimension $d_h \in \{2, 4, 8, 16, 32, 64, 96, 192\}$. The trainable weights are i.i.d. initialized using a standard normal distribution $\mathcal{N}(0, 1)$. We also generate random matrices with i.i.d. entries following $\mathcal{N}(0, 1)$ with a shape of $(n, b, d)$, where the sequence length $n$ is set as 100, the batch size $b$ is 32 and the data dimension $d$ is 384. Subsequently, we record the mean and standard deviation of all attention matrices for every $d_h$.

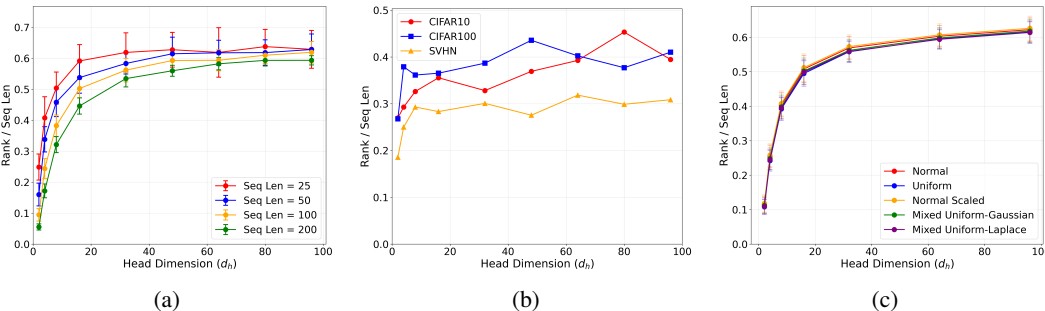

(a)                                    (b)                                    (c)

Figure 2: The consistent pattern of attention ranks across varied experimental conditions: (a) different sequence lengths (25, 50, 100 and 200); (b) different real-world datasets (CIFAR-10/100 (Krizhevsky et al., 2009), and SVHN (Netzer et al., 2011)); (c) different types of (synthetic) data distributions and non-i.i.d. cases.

**Rank Calculation.** There are several equivalent definitions of the matrix rank in algebra. For numerical computation, the rank is usually calculated via singular value decomposition (SVD), i.e., the rank equals to the number of non-zero singular values. In practice, due to the numerical precision limitation and round-off errors, this procedure often requires a relaxation, where a tolerance threshold $\epsilon$ is applied to yield the so-called numerical matrix rank. That is, $\mathrm{rank}(\mathbf{A}, \epsilon)$ equals to the number of singular values no less than $\epsilon$. Here, we set the tolerance threshold as $\epsilon = 10^{-8}$.

**Observations.** The experimental results (which is visualized in Figure 1) illustrate a clear relationship between the head dimension $d_h$ and Rank / Seq Len. For relatively small values of $d_h$, the attention matrix exhibits a low rank, which increases normally as $d_h$ increases (i.e. successive increases in ranks are relatively large from $d_h = 2$ to $d_h = 16$). However, for appropriately large values of $d_h$, further increases in $d_h$ lead to *diminishing increments* of attention ranks, with a final barrier of approximately $0.63n \ll n$ ($n$: the full-rank). This diminishing return pattern is evident in the data: While Rank / Seq Len increases by around 0.10 from $d_h = 8$ to $d_h = 16$, as $d_h$ further rises to 192, the increment in Rank / Seq Len reduces to around 0.01, suggesting a more significant plateauing effect at higher $d_h$ levels. Additionally, the variances in Rank / Seq Len exhibit slight fluctuations across different $d_h$ values but remain relatively low, demonstrating the stability of experimental results. The observations are summarized as follows.

- The attention rank increases with the head dimension $d_h$. When $d_h$ increases within relatively small values, there is a notable rise in the attention rank.

- When $d_h$ is appropriately large, further increases in $d_h$ result in only marginal increments of attention ranks, which is capped at around $0.63n \ll n$ (the full-rank).

## 2.2 ABLATION STUDIES ON DATASETS

**Sequence Lengths.** We examine the influence of sequence lengths on attention ranks by varying lengths in $\{25, 50, 100, 200\}$. To ensure a comprehensive investigation, we test a refined set of head dimensions ($d_h \in \{2, 4, 8, 16, 32, 48, 64, 80, 96\}$) and increase the model dimension to $d_{\mathrm{model}} = 960$. The other configurations remain the same as those outlined in Section 2.1. The results summarized in Table 1 and Figure 2a show the ratio of attention ranks over sequence lengths (Rank/Seq Len) versus head dimensions ($d_h$) for different sequence lengths. Despite of varied sequence lengths, all curves exhibit consistent patterns: attention ranks increase with head dimensions but eventually saturate at approximately $0.63n$. Notably, as highlighted in Table 1, the required head dimensions for the saturation of attention ranks exhibit a linear increase with doubling sequence lengths, with saturation points occurring at progressively larger head dimensions. This suggests a logarithmic dependency ($d_h = \Omega(\log n)$) aligned with by our theoretical analysis (Section 3.2), further confirming the robustness of our findings in Section 2.1.

**Real-World Datasets.** In Figure 2b, we show that the above findings (in Section 2.1) that attention ranks are capped and get saturated are consistent across diverse visual recognition tasks, including

CIFAR-10, CIFAR-100, and SVHN datasets. Despite of different characteristics and complexities of theses datasets, similar curves of attention ranks versus head dimensions are observed, further validating the generalizability of our findings.

**Data Distributions.** We also investigate attention ranks for different types of (synthetic) data distributions with scales, including $\mathcal{N}(0, 1)$, $\mathcal{N}(0, 100)$, $\mathcal{U}(-1, 1)$ and $\mathcal{U}(-100, 100)$, and consistent phenomena irrespective of distributions are observed. For comprehensive discussions and detailed experimental reports, refer to Appendix B.4. Figure 2c shows that similar patterns hold for various non-i.i.d. and mixed distributions. The rand_randn line represents tensors where half of the elements are sampled from a uniform distribution and the other half from a Gaussian distribution, while the rand_double_exponential line denotes tensors where half of the elements are sampled from a uniform distribution and the other half from a double exponential distribution. These results verify the generalizability of attention rank patterns across diverse data conditions, underscoring the robustness of our findings w.r.t. data distributions.

Table 1: Attention ranks versus sequence lengths. The highlighted boldface statistics are set according to the "Improvement" column: when the improvement drops less than or around 0.01 for the first time at a certain row, we set the *above* one row as the critical position of $d_h$ where the saturation of attention ranks begins to occur. One can observe that as the sequence length doubles, the required head dimension to reach the saturation increases linearly, potentially implying certain log-dependence.

| $d_h$ | Seq Len = 25 | | Seq Len = 50 | | Seq Len = 100 | | Seq Len = 200 | |
|---|---|---|---|---|---|---|---|---|
| | Rank/Seq Len | Improvement | Rank/Seq Len | Improvement | Rank/Seq Len | Improvement | Rank/Seq Len | Improvement |
| 2 | $0.250 \pm 0.051$ | - | $0.158 \pm 0.029$ | - | $0.096 \pm 0.019$ | - | $0.055 \pm 0.011$ | - |
| 4 | $0.422 \pm 0.061$ | +0.172 | $0.324 \pm 0.044$ | +0.166 | $0.240 \pm 0.032$ | +0.144 | $0.172 \pm 0.019$ | +0.117 |
| 8 | $0.530 \pm 0.068$ | +0.108 | $0.459 \pm 0.047$ | +0.135 | $0.391 \pm 0.035$ | +0.151 | $0.323 \pm 0.025$ | +0.151 |
| 16 | $\mathbf{0.606 \pm 0.055}$ | +0.076 | $0.536 \pm 0.052$ | +0.077 | $0.498 \pm 0.029$ | +0.107 | $0.443 \pm 0.026$ | +0.120 |
| 32 | $0.612 \pm 0.066$ | +0.006 | $\mathbf{0.593 \pm 0.045}$ | +0.057 | $0.571 \pm 0.031$ | +0.073 | $0.525 \pm 0.023$ | +0.082 |
| 48 | $0.618 \pm 0.048$ | +0.006 | $0.601 \pm 0.033$ | +0.008 | $\mathbf{0.594 \pm 0.034}$ | +0.023 | $0.554 \pm 0.018$ | +0.029 |
| 64 | $0.621 \pm 0.060$ | +0.003 | $0.612 \pm 0.057$ | +0.011 | $0.606 \pm 0.038$ | +0.012 | $\mathbf{0.579 \pm 0.021}$ | +0.025 |
| 80 | $0.623 \pm 0.071$ | +0.002 | $0.615 \pm 0.054$ | +0.003 | $0.609 \pm 0.049$ | +0.003 | $0.592 \pm 0.018$ | +0.013 |
| 96 | $0.625 \pm 0.058$ | +0.002 | $0.622 \pm 0.058$ | +0.007 | $0.611 \pm 0.034$ | +0.002 | $0.597 \pm 0.020$ | +0.005 |

## 2.3 ABLATION STUDIES ON HYPERPARAMETERS

**Model Dimensions.** We first investigate the effect of different model dimensions $d_{\text{model}} \in \{384, 768, 1152, 1536\}$, maintaining other configurations specified in Section 2.1. The results (provided in Appendix B.1) align with Figure 1, indicating a robust and consistent pattern of attention ranks across varied model dimensions.

**Softmax Temperatures.** We test the softmax temperature $T \in \{10^{-5}, 10^{-3}, 10^{-1}, 1\}$ to assess its effect on the attention rank. Similarly, the outcomes (detailed in Appendix B.2) also exhibit a robust and consistent pattern of attention ranks across different softmax temperatures.

**Transformers' Layers.** To study the attention ranks in different layers, we test a 8-layer Transformer. The results (elaborated in Appendix B.3) also similarly reveal a consistent pattern among different layers.

## 3 THEORETICAL ANALYSIS

In this section, we provide the fine-grained mathematical analysis to demonstrate rigorously the experimental results reported in Section 2, i.e. the existence of the low-rank barrier and rank-saturation effect.

### 3.1 MAIN RESULTS

Our goal to theoretically characterize the low-rank barrier and rank-saturation effect can be formulated as follows. That is, (i) there exists a non-trivial upper bound ($\approx 0.63n$) of the attention rank (i.e.

$\text{rank}\left(\mathbf{Attn}(\mathbf{X};\boldsymbol{\theta}))\right)$ in expectation regardless of the head dimension $d_h$; (ii) $\text{rank}\left(\mathbf{Attn}(\mathbf{X};\boldsymbol{\theta})\right)$ gets saturated when $d_h = \Omega(\log n)$.

For convenience, we focus on the low-temperature case (i.e. $T > 0$ appropriately small) associated with the "hardmax" activation. Note that although we employ this setup for theoretical simplicity, the hardmax activation is occasionally used in applications for computational efficiency. See computer vision (CV) examples in (Elsayed et al., 2019; Papadopoulos et al., 2021) for more details. When $T > 0$ is appropriately small, it holds that

$$\text{softmax}\left(\frac{\mathbf{X}\mathbf{W}_q\mathbf{W}_k^\top\mathbf{X}^\top}{T}\right) \approx \text{hardmax}\left(\mathbf{X}\mathbf{W}_q\mathbf{W}_k^\top\mathbf{X}^\top\right), \tag{1}$$

where the maximum is taken in a row-wise sense: for a matrix $\mathbf{A} = [a_{ij}] \in \mathbb{R}^{n \times n}$, $\mathbf{e}_i^\top\text{hardmax}(\mathbf{A}) := \mathbf{e}_{k_i}$ with $k_i := \arg\max_{j\in[n]} a_{ij}$.

**Remark 1.** *Numerically, we have demonstrated in Figure 5b that the attention rank of Transformers is robust to variations in softmax temperatures, as least in the range between low temperatures (hardmax) and normal temperatures (softmax). In this work, all the experiments are performed for normal temperatures, obtaining results consistent with the following theory.*

We have the following main theorem to estimate the (averaged) rank of (1). The derived upper bound (proofs deferred to Appendix A) coincides perfectly with the experimental results in Figure 1.

**Theorem 1.** *Let the parameters $\mathbf{W}_q, \mathbf{W}_k$ be Gaussian random matrices, i.e., the entries of $\mathbf{W}_q, \mathbf{W}_k$ are independent $\mathcal{N}(0,1)$ random variables. Assume that the input sequence $\mathbf{X}$ satisfies $\mathbf{X}\mathbf{X}^\top = \mathbf{I}_n + \mathbf{E}$ with $\mathbf{E} = [E_{ij}] \in \mathbb{R}^{n\times n}$ satisfying $|E_{ij}| \leq \epsilon = o(1/(n^{\frac{3}{2}}(d + d_h)))$ ($\forall i,j \in [n]$, i.e. almost orthonormality of inputs). Then for any $n \in \mathbb{N}_+$ appropriately large, $d \geq n$, and $\delta > 0$ appropriately small, we have*

$$\mathbb{E}_{\mathbf{W}_k,\mathbf{W}_q}\left[\text{rank}\left(\text{hardmax}\left(\mathbf{X}\mathbf{W}_q\mathbf{W}_k^\top\mathbf{X}^\top\right),\delta\right)\right]$$
$$\leq (1 - \exp(-1))n + O(1) \approx 0.63n, \tag{2}$$

*where $\text{rank}(\mathbf{A},\delta)$ equals to the number of singular values (of $\mathbf{A}$) no less than $\delta$ (i.e. numerical rank). Furthermore, the left hand side of (2) is approximately independent of the head dimension $d_h$ when $d_h = \Omega(\log n)$.*

The proof of Theorem 1 is deferred to Appendix A. It is worthwhile to note that almost orthonormality leads to *exponentially* many "basis" vectors (rather than *linear* for exact orthonormality) owing to Johnson–Lindenstrauss lemma.

**Remark 2.** *The assumption that the input sequence is almost orthonormal might seem stringent at the first glance. However, in practical scenarios, particularly in high-dimensional spaces ($d \gg 1$), the (embedding) vectors (i.e. $\mathbf{x}_i$ here) representing different tokens can be almost orthogonal, if they are modeled using independent and isotropic Gaussian random vectors (Vershynin, 2018). This assumption is also proposed by Tian et al. (2024) to theoretically analyze the training dynamics of Transformers. According to Tian et al. (2024), the almost orthogonality even holds during the training process (for large pre-trained models such as Pythia, BERT, OPT, LLaMA and ViT of different sizes). We also numerically verify the orthonormality by ourselves in Appendix B.5 (Figure 6) on both synthetic and real-world datasets.*

**Remark 3.** *Note that the $\text{hardmax}$ operator is invariant under the positive scaling: $\text{hardmax}(c\mathbf{A}) = \text{hardmax}(\mathbf{A})$ for any $c > 0$. Consequently, Theorem 1 remains valid even in cases where input sequences are not normalized.*

**Low-Rank Bottleneck on Approximation.** According to Eckart–Young theorem (Eckart & Young, 1936), there exists a lower bound corresponding to the spectral regularity, a.k.a. low-rank approximation problem. For instance, given the target matrix $\mathbf{A} \in \mathbb{R}^{n\times n}$ with singular values $\sigma_1 \geq \cdots \geq \sigma_{n'} > \sigma_{n'+1} = \cdots = \sigma_n = 0$ (i.e. $\text{rank}(\mathbf{A}) = n' \in (0.63n, n]$), based on Eckart–Young theorem and Theorem 1, we have $\left\|\text{hardmax}\left(\mathbf{Q}\mathbf{K}^\top\right) - \mathbf{A}\right\|_F^2 \geq \sum_{i=\text{rank}(\text{hardmax}(\mathbf{Q}\mathbf{K}^\top))+1}^{n'} \sigma_i^2 \overset{\text{e}}{\geq}$

$\sum_{i=(1-\exp(-1))n+O(1)}^{n'} \sigma_i^2 \approx \sum_{i=0.63n}^{n'} \sigma_i^2 > 0$ for any $n \in \mathbb{N}_+$ appropriately large, where $\overset{\text{e}}{\geq}$ represents "no less than" in expectation. One can expect that this lower bound implies a large gap of low-rank approximation if the spectrum of $\mathbf{A}$ (i.e. $\{\sigma_i\}_{i=1}^n$) decays slowly (e.g. $\mathbf{A}$ has a full rank $n$).

## 3.2 DISCUSSIONS

In this section, we revisit the experimental results in Section 2, and compare them with the developed theoretical results in Section 3.1. Comparing the estimate (2) and the bound $d_h = \Omega(\log n)$ in Theorem 1 with the observations in Section 2, we obtain the *consistency* between our theoretical results and simulation outcomes.

First, considering Figure 1 (and Figure 5a, 5b, 5c) and Table 1 (and Table 2), we note that under various settings (such as different model dimensions, softmax temperatures, model depths, sequence lengths and data distributions), the attention rank increases with the head dimension $d_h$, yet it converges towards the upper bound predicted by the estimate (2). Furthermore, the incremental growth of the attention rank significantly diminishes with a uniform increase in $d_h$, indicating an obvious trend towards the saturation.

Second, we focus on Table 1. Based on the highlighted boldface statistics, it is evident that for *doubled* sequence lengths, a distinct *linear increment* trend of head dimensions for rank saturation is observed. For instance, at the sequence length of $n = 25$, the saturation occurs at $d_h = 16$; for sequence lengths of $n = 50, 100, 200$, the critical saturation positions are identified at $d_h = 32, 48$ and 64, respectively. This finding quantitatively aligns with the theoretical estimate $d_h = \Omega(\log n)$.

## 4 REAL-WORLD EXPERIMENTS: MODEL-REDUCTION

In this section, we further verify our previous findings through simulations on real-world datasets. In theory, the upper bound is derived for every single head. For the multiple heads case, we aim to emphasize the *saturation* or model-reduction effect via numerical simulations. That is, despite that one can increase the overall rank by concatenation in multiple-head attention, the low-rank saturation of every single head still leads to an *inefficiency* issue: As is shown later, both the attention rank and model performance *consistently* get *marginal enhancements* when increasing parameters, implying the principled model redundancy. This gives chances for the optimal configuration of hyper-parameters: In practical applications, one may check the saturation situation of attention ranks before training, and set the optimal number of parameters as where the rank first gets saturated.

### 4.1 REAL-WORLD EXPERIMENTS ON NLP TASKS

The experiments focus on evaluating the performance of Transformers on text classification tasks using the IMDB dataset (Maas et al., 2011). In this section, we fix the number of heads, and then vary the head dimension $d_h \in \{2, 3, 4, 8, 16\}$, which deviates from the conventional constraint $d = h \times d_h$. With this configuration, we can directly observe the relationship between head dimensions, and both model performance and attention rank saturation:

1. In Figure 3(a), it is shown that the learning accuracy increases significantly as $d_h$ grows within relatively small values. However, this improvement plateaus once $d_h$ becomes appropriately large, reflecting diminishing marginal returns with further parameter expansions. The optimal configuration occurs at $d_h^* = 8$ (right before the marginal improvement).

2. Notably, the corresponding attention ranks[1] in Figure 3(b) exhibit similar saturation behaviors when $d_h \geq d_h^* = 8$, which aligns with the saturated trends of learning performance observed in Figure 3(a). This correlation between attention rank saturation and performance plateauing validates our theoretical analysis of the model-reduction effect in practice.

3. To further study the effect of input sizes and Transformer layers on attention ranks, we examine rank saturation at different Transformer layers for varied embedding dimensions within $\{32, 128, 256, 512\}$ on the IMDB dataset. Figures 3(c) and 3(d) show the experimental results for the first and second layers, respectively. The results consistently demonstrate that rank saturation appears across different Transformer layers as the input embedding dimension varies, reinforcing our findings on the fundamental nature of model-reduction.

---

[1]The ranks in Figure 3(b) are calculated for the first-layer attention matrices at initialization, computed on mini-batches of IMDB tokens and averaged over multiple runs with varied random seeds.

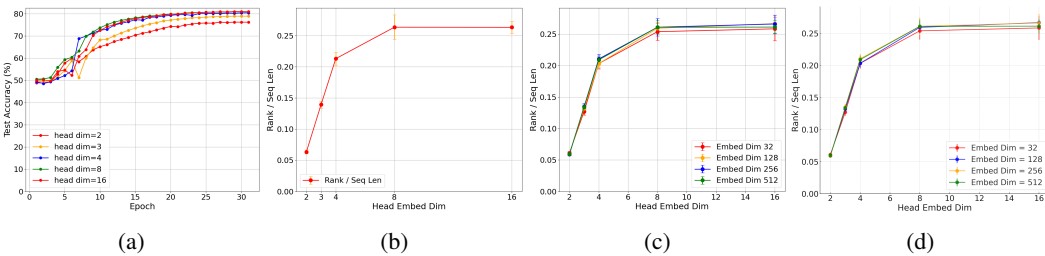

(a)  (b)  (c)  (d)

Figure 3: Real-world experiments on the IMDB dataset for varied head dimensions (with the number of heads fixed). (a), (b): both learning performance and attention ranks consistently get diminishing returns; (c), (d): rank saturation across varied embedding dimensions at different Transformer layers.

**Remark 4.** *The findings necessitate and support the usage of multi-head* latent *attention (MLA; (Liu et al., 2024a)) that applies low-rank input embeddings corresponding to relatively small head dimensions. This approach has been successfully verified to reduce memory usage while maintaining performance in DeepSeek-V3 (Liu et al., 2024b), thereby enhancing the modeling and learning efficiency.*

## 4.2 REAL-WORLD EXPERIMENTS ON CV TASKS

The experiments focus on evaluating the performance of Vision Transformers (ViTs; (Dosovitskiy et al., 2021)) on image classification tasks using the CIFAR-10 dataset.

To include more cases, here we instead fix the model dimension $d_{\text{model}} = d$, and vary the number of heads $h$ (and consequently the head dimension $d_h$) following the equation $d = h \times d_h$, which is default in practical applications.

The model-reduction based explanation can be as follows. With the above constraint, a smaller number of heads $h$ results in a larger head dimension $d_h$, potentially exceeding the critical head dimension to achieve the rank saturation for each head. Namely, most of the heads may have reached the saturation point, leading to the redundancy in modeling parameters. On the contrary, as the number of heads increases, the Transformer model with reduced head dimensions gradually avoids rank saturation (and potential parameter redundancy), leading to more portions of "effective" ranks for modeling, which yields improved experimental results.

These arguments are numerically supported by jointly examining Figure 4a and Figure 4b. Figure 4a shows that increasing the number of heads ($h = 1, 2, 4, 8$) benefits the model's performance in general, while the attention ranks[2] get saturated at the corresponding head dimension $d_h = 384, 192, 96, 48$ ($d_{\text{model}} = 384$) in Figure 4b. The results show that under these configurations, the saturated attention ranks lead to the fact that appropriately decreasing $d_h$ will not affect the expressive ability of each head, and the model performance will instead improve from an increase in the number of heads. For experiments on more datasets and the head-fixed regime (similar to Section 4.1), see Appendix C.2 and Appendix C.3 for details.

## 5 RELATED WORK

The rank of attention matrices in Transformers has attracted extensive research (Kanai et al., 2018; Bhojanapalli et al., 2020; Dong et al., 2021; Lin et al., 2022). Bhojanapalli et al. (2020) identified a restriction from the low-rank bottleneck in attention heads, showing that low-rank attention cannot capture certain contexts. They attributed this to the proportional relationship between the number of heads and head size in standard architectures. Dong et al. (2021) offered a new perspective on self-attention networks, demonstrating that without skip connections and multi-layer perceptrons (MLPs), outputs quickly degenerate to a rank-1 matrix, causing pure attention to lose expressive power exponentially with depth.

---

[2]The ranks in Figure 4b are calculated for the first-layer attention matrices on a mini-batch of CIFAR-10 images, averaged over both all heads and multiple varied random seeds.

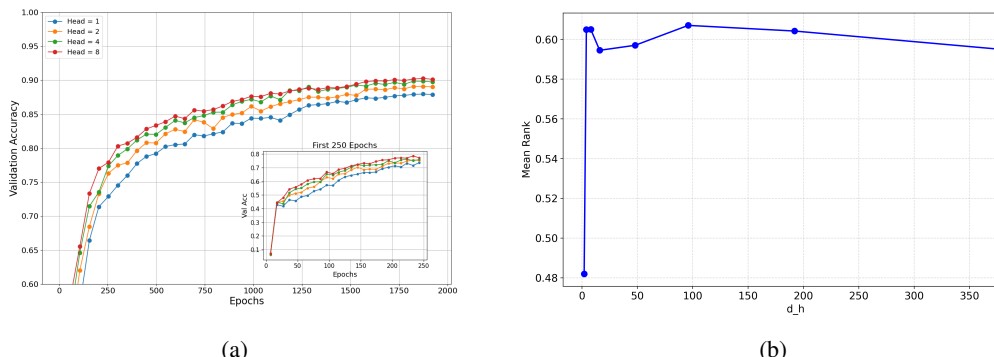

(a)                                                        (b)

Figure 4: Real-world experiments on the CIFAR-10 dataset for varied number of heads (with model dimensions fixed). (a): model performance improves as the number of heads increases; (b): attention ranks get saturated. The results show that as the number of heads increases, Transformers with reduced head dimensions gradually avoid rank saturation, leading to more portions of "effective" ranks for modeling and hence improved performance.

Meanwhile, Transformer variants have sought to overcome computational and memory bottlenecks (Chen et al., 2021a; Wang et al., 2020; Hu et al., 2022; Guo et al., 2019; Lin et al., 2022). For example, Wang et al. (2020) showed that self-attention complexity can be reduced using low-rank approximations. Guo et al. (2019) imposed low-rank constraints that improved performance on certain tasks. Chen et al. (2021a) reported that sparse and low-rank approximations are effective under different conditions, with combined approaches outperforming either method alone.

Another direction focuses on computational efficiency, such as KDEformer (Zandieh et al., 2023) and HyperAttention (Han et al., 2024). These methods approximate attention matrices by replacing full multiplications with smaller sub-matrix operations, where ranks depend on matrix spectra. Future work may extend these ideas using the inductive biases identified here, to design more efficient algorithms under the low-rank barrier and rank saturation.

Compared with these studies, our work investigates the ranks of attention score matrices in Transformers and provides two insights: attention rank increases with head dimension but has an upper limit (*low-rank barrier*), and a *model-reduction effect* emerges. These findings are consistently validated across models and datasets, and supported by theoretical analysis.

## 6 CONCLUSION

In this work, we conduct a comprehensive study of the rank of attention matrices in Transformers, combining theoretical analysis with empirical evidence. Theoretically, we establish a strict upper bound on attention rank that is significantly lower than full rank, indicating the presence of a low-rank barrier. We also show that when head dimensions are small relative to sequence length, the attention rank saturates, suggesting that further parameter increases yield diminishing performance gains (model-reduction effect).

Experimentally, we validate these findings through extensive simulations across diverse model architectures and real-world datasets. The results confirm the robustness of our theory in practical settings. The identified relationship between head dimensions, attention rank, and model performance offers a clearer understanding of Transformer models' capacity and efficiency.

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

# A    PROOFS

In this section, we provide all the missing proofs. To prove the main theorem (Theorem 1), we first analyze the setting where input sequences are exactly orthonormal (Section A.1). Then, we extend the above analysis to the almost orthonormality setting via approximation procedures and stability/perturbation analysis (Section A.2).

## A.1    ANALYSIS UNDER ORTHONORMALITY

The proof entails a detailed analysis of matrix operations, probability transforms, and infinitesimal order estimation. Specifically, the proof sketch proceeds as follows:

- First, given the orthonormal nature of input sequences, according to Lemma 4, one can derive that different rows of $\mathbf{X}\mathbf{W}_q\mathbf{W}_k^\top\mathbf{X}^\top$ are independent, and these rows are identically distributed as $\mathcal{N}(\mathbf{0}_n, \mathbf{K}\mathbf{K}^\top)$, conditioned on any fixed Gaussian random matrix $\mathbf{W}_k$.

- Then, note that applying the hardmax operation to individual rows is analogous to solving an elementary birthday problem (refer to Lemma 3), which reduces the original problem as counting columns with all zeros.

- Finally, the estimate is further refined based on Lemma 2, and completed by applying the AM-GM inequality, which indicates the equality when all probabilities are equal.

To begin with, the key approximation (1) is due to the following lemma, which characterizes the gap between the softmax function and its "hard" version.

**Lemma 1.** *Let* $\mathbf{a} = [a_1, a_2, \cdots, a_n]^\top \in \mathbb{R}^n$ *with* $i^* := \arg\max_{i \in [n]} a_i$ *and* $i'^* := \arg\max_{i \in [n], i \neq i^*} a_i$, *and* $\mathrm{hardmax}(\mathbf{a}) := \mathbf{e}_{i^*}$. *Assume that* $\delta := a_{i^*} - a_{i'^*} > 0$ *(i.e., the maximum is unique). Then for any* $T > 0$, *we have*

$$\begin{aligned}\Delta_{n,\delta}(T) &:= \|\mathrm{softmax}(\mathbf{a}/T) - \mathrm{hardmax}(\mathbf{a})\|_1 \\ &\leq 2(n-1)\exp(-\delta/T).\end{aligned} \tag{3}$$

*That is,* $\Delta_{n,\delta}(T)$ *converges to 0 exponentially fast as* $T \to 0^+$.

*Proof.* It is straightforward to have

$$\Delta_{n,\delta}(T) = \sum_{i \in [n], i \neq i^*} \frac{\exp(a_i/T)}{\sum_{j=1}^n \exp(a_j/T)}$$

$$+ 1 - \frac{\exp(a_{i^*}/T)}{\sum_{j=1}^n \exp(a_j/T)}$$

$$= 2 \frac{\sum_{i \in [n], i \neq i^*} \exp(a_i/T)}{\sum_{i \in [n], i \neq i^*} \exp(a_i/T) + \exp(a_{i^*}/T)}$$

$$\leq 2 \sum_{i \in [n], i \neq i^*} \exp((a_i - a_{i^*})/T)$$

$$\leq 2(n-1) \exp((a_{i'^*} - a_{i^*})/T)$$

$$= 2(n-1) \exp(-\delta/T). \tag{4}$$

This gives $\lim_{T \to 0^+} \Delta_{n,\delta}(T) = 0$, and the rate is exponentially fast. The proof is completed. □

Before we prove the low-rank barrier and model-reduction effect of (1), the following lemmas are useful.

**Lemma 2.** *For any $n \in \mathbb{N}_+$, define $\delta_n(p) := \exp(-pn) - (1-p)^n$, $p \in [0, +\infty)$. Then we have*

$$\delta_n(p) \leq \frac{1}{2} p^2 n \exp(-p(n-1)) \tag{5}$$

$$\leq \begin{cases} \frac{1}{2} p^2, & n = 1, \\ 2 \exp(-2) \left( \frac{1}{n-1} + \frac{1}{(n-1)^2} \right), & n \geq 2. \end{cases} \tag{6}$$

*Proof.* Note that $a_1^n - a_2^n = (a_1 - a_2) \sum_{k=0}^{n-1} a_1^{n-1-k} a_2^k$ for any $a_1, a_2 \in \mathbb{R}$, we have

$$\delta_n(p) = (\exp(-p))^n - (1-p)^n \tag{7}$$

$$= [\exp(-p) - (1-p)]$$

$$\times \sum_{k=0}^{n-1} (\exp(-p))^{n-1-k} (1-p)^k. \tag{8}$$

Let $g_1(p) := \exp(-p) - (1-p)$, $g_2(p) := \exp(-p) - (1-p+p^2/2) = g_1(p) - p^2/2$, $p \in [0, +\infty)$, we get

$$g_1'(p) = -\exp(-p) + 1 \geq 0 \tag{9}$$

$$\Rightarrow g_1(p) \geq g_1(0) = 0, \tag{10}$$

$$g_2'(p) = -\exp(-p) + 1 - p = -g_1(p) \leq 0 \tag{11}$$

$$\Rightarrow g_2(p) \leq g_2(0) = 0, \tag{12}$$

which gives

$$\delta_1(p) = g_1(p) \leq p^2/2, \tag{13}$$

$$\delta_n(p) \leq \frac{1}{2} p^2 \sum_{k=0}^{n-1} (\exp(-p))^{n-1-k} (\exp(-p))^k \tag{14}$$

$$= \frac{1}{2} p^2 n (\exp(-p))^{n-1}, \quad n \geq 2. \tag{15}$$

For any $n \in \mathbb{N}_+$, $n \geq 2$, let $h_n(p) := p^2 (\exp(-p))^{n-1}$, $p \in [0, +\infty)$, we get $h_n'(p) = p(\exp(-p))^{n-1}(2 - p(n-1))$, hence

$$h_n'(p) = 0 \Rightarrow p = 0 \text{ or } p = 2/(n-1) \tag{16}$$

$$\Rightarrow h_n(p) \leq h_n(2/(n-1)) \tag{17}$$

$$= \frac{4 \exp(-2)}{(n-1)^2}. \tag{18}$$

Therefore, for $n \geq 2$, we obtain

$$\delta_n(p) \leq \frac{1}{2} n h_n(p) \tag{19}$$

$$\leq \frac{2 \exp(-2) n}{(n-1)^2} \tag{20}$$

$$= 2 \exp(-2) \left( \frac{1}{n-1} + \frac{1}{(n-1)^2} \right), \tag{21}$$

which completes the proof. $\qquad\square$

**Lemma 3.** *For a random matrix $\mathbf{A} = [a_{ij}] \in \mathbb{R}^{n \times n}$ with independent rows, let $p_{ij} := \mathbb{P}(\{a_{ij} = \max_{j' \in [n]} a_{ij'}\})$. Then the expectation number of columns with all zeros in $\mathrm{hardmax}(\mathbf{A})$ is*

$$\sum_{j=1}^{n} \prod_{i=1}^{n} (1 - p_{ij}). \tag{22}$$

*Proof.* For $j = 1, 2, \ldots, n$, define the random variable

$$X_j = \begin{cases} 1, & \mathrm{hardmax}(\mathbf{A})\mathbf{e}_j = \mathbf{0}_n, \\ 0, & \mathrm{hardmax}(\mathbf{A})\mathbf{e}_j \neq \mathbf{0}_n. \end{cases} \tag{23}$$

By independence, we get

$$\mathbb{P}(\{X_j = 1\}) = \mathbb{P}\left( \bigcap_{i=1}^{n} \{\mathbf{e}_i^\top \mathrm{hardmax}(\mathbf{A})\mathbf{e}_j = 0\} \right)$$

$$= \prod_{i=1}^{n} \mathbb{P}\left( \{\mathbf{e}_i^\top \mathrm{hardmax}(\mathbf{A})\mathbf{e}_j = 0\} \right)$$

$$= \prod_{i=1}^{n} (1 - p_{ij}). \tag{24}$$

Therefore, the expectation number of columns with all zeros is

$$\mathbb{E}\left[ \sum_{j=1}^{n} X_j \right] = \sum_{j=1}^{n} \mathbb{E}[X_j] \tag{25}$$

$$= \sum_{j=1}^{n} \mathbb{P}(\{X_j = 1\}) \tag{26}$$

$$= \sum_{j=1}^{n} \prod_{i=1}^{n} (1 - p_{ij}), \tag{27}$$

which completes the proof. $\qquad\square$

The required independence in Lemma 3 is provided by the following lemma.

**Lemma 4.** *((Vershynin, 2018), Exercise 3.3.6) Let $\mathbf{G} \in \mathbb{R}^{m \times n}$ be a Gaussian random matrix, i.e. the entries of $\mathbf{G}$ are independent $\mathcal{N}(0,1)$ random variables. Let $\mathbf{u}, \mathbf{v} \in \mathbb{R}^n$ be unit orthogonal vectors. Then, $\mathbf{Gu}$ and $\mathbf{Gv}$ are independent $\mathcal{N}(\mathbf{0}_m, \mathbf{I}_m)$ random vectors.*

*Proof.* First, we show that $\mathbf{Gu}, \mathbf{Gv}$ are both $\mathcal{N}(\mathbf{0}_m, \mathbf{I}_m)$ random vectors. This is straightforward since $\mathbf{Ge}_j \sim \mathcal{N}(\mathbf{0}_m, \mathbf{I}_m)$ gives $u_j \mathbf{Ge}_j \sim \mathcal{N}(\mathbf{0}_m, u_j^2 \mathbf{I}_m)$, and $\{u_j \mathbf{Ge}_j\}_{j=1}^n$ is a collection of independent Gaussian vectors. Hence $\mathbf{Gu} = \sum_{j=1}^{n} u_j \mathbf{Ge}_j \sim \mathcal{N}(\mathbf{0}_m, \|\mathbf{u}\|_2^2 \mathbf{I}_m)$.

Next, we show the independence of $\mathbf{Gu}$ and $\mathbf{Gv}$. Equivalently, we are supposed to prove that $\mathbf{e}_i^\top \mathbf{Gu}$ and $\mathbf{e}_{i'}^\top \mathbf{Gv}$ are independent random variables for any $i, i' \in [n]$. For $i \neq i'$, $(\mathbf{e}_i^\top \mathbf{G})\mathbf{u}$ and $(\mathbf{e}_{i'}^\top \mathbf{G})\mathbf{v}$

are independent random variables since $\mathbf{G}$ has independent rows. Therefore, the problem is reduced as the independence of $\mathbf{g}^\top \mathbf{u}$ and $\mathbf{g}^\top \mathbf{v}$ for $\mathbf{g} \sim \mathcal{N}(\mathbf{0}_n, \mathbf{I}_n)$. Notice that

$$[\mathbf{u}, \mathbf{v}]^\top \mathbf{g} \sim \mathcal{N}(\mathbf{0}_2, [\mathbf{u}, \mathbf{v}]^\top \mathbf{I}_n [\mathbf{u}, \mathbf{v}]) \tag{28}$$
$$= \mathcal{N}(\mathbf{0}_2, \mathbf{I}_2), \tag{29}$$

which completes the proof. $\qquad\square$

Now we are ready to prove the main theorem given the exact orthonormality condition.

**Theorem 2.** *(Theorem 1 under orthonormality) Let the parameters* $\mathbf{W}_q, \mathbf{W}_k$ *be Gaussian random matrices, i.e., the entries of* $\mathbf{W}_q, \mathbf{W}_k$ *are independent* $\mathcal{N}(0,1)$ *random variables. Assume that the input sequence* $\mathbf{X}$ *satisfies* $\mathbf{X}\mathbf{X}^\top = \mathbf{I}_n$. *Then for any* $n \in \mathbb{N}_+$, $n \geq 2$, *we have*

$$\mathbb{E}_{\mathbf{W}_k, \mathbf{W}_q} \left[ \mathrm{rank}\left( \mathrm{hardmax}\left( \mathbf{X}\mathbf{W}_q \mathbf{W}_k^\top \mathbf{X}^\top \right) \right) \right] \tag{30}$$
$$\leq (1 - \exp(-1))n + 2\exp(-2)[1 + 1/(n-1)]^2 \tag{31}$$
$$\approx (1 - \exp(-1))n \tag{32}$$
$$\approx 0.63n, \quad n \text{ appropriately large}. \tag{33}$$

*Proof.* According to Lemma 4, since $\mathbf{x}_i^\top \mathbf{x}_j = \delta_{ij}$ (Kronecker symbol), $i, j = 1, 2, \cdots, n$, one can deduce that $\{\mathbf{q}_i\}_{i=1}^n = \{\mathbf{W}_q^\top \mathbf{x}_i\}_{i=1}^n$ is a collection of independent $\mathcal{N}(\mathbf{0}_{d_h}, \mathbf{I}_{d_h})$ random vectors. For any fixed Gaussian random matrix $\mathbf{W}_k$,

$$(\mathbf{e}_i^\top \mathbf{X}\mathbf{W}_q \mathbf{W}_k^\top \mathbf{X}^\top)^\top = \mathbf{K}\mathbf{q}_i \sim \mathcal{N}(\mathbf{0}_n, \mathbf{K}\mathbf{K}^\top), \tag{34}$$

which is also independent across different $i$'s. That is to say, the rows of $\mathbf{X}\mathbf{W}_q \mathbf{W}_k^\top \mathbf{X}^\top$ are independent and identically distributed as $\mathcal{N}(\mathbf{0}_n, \mathbf{K}\mathbf{K}^\top)$. Therefore, according to Lemma 3, the expectation number of columns with all zeros in $\mathrm{hardmax}(\mathbf{X}\mathbf{W}_q \mathbf{W}_k^\top \mathbf{X}^\top)$ is

$$\sum_{j=1}^n \prod_{i=1}^n (1 - p_{ij}) = \sum_{j=1}^n \prod_{i=1}^n (1 - p_j) \tag{35}$$
$$= \sum_{j=1}^n (1 - p_j)^n. \tag{36}$$

Hence, we have

$$\frac{1}{n}\mathbb{E}_{\mathbf{W}_q} \left[ \mathrm{rank}\left( \mathrm{hardmax}\left( \mathbf{X}\mathbf{W}_q \mathbf{W}_k^\top \mathbf{X}^\top \right) \right) \right]$$
$$\leq 1 - \frac{1}{n} \sum_{j=1}^n (1 - p_j)^n. \tag{37}$$

Note that $[p_1, p_2, \cdots, p_n]$ is a probability vector, i.e. $\sum_{j=1}^n p_j = 1$, $p_j \geq 0$ for any $j \in [n]$, and $\exp(-p) \geq 1 - p \geq 0$ for any $p \in [0, 1]$, we get $\delta_n(p) = \exp(-pn) - (1-p)^n \geq 0$ for any $p \in [0, 1]$. Therefore, by Lemma 2, we have

$$\frac{1}{n} \sum_{j=1}^n |(1 - p_j)^n - \exp(-p_j n)|$$
$$= \frac{1}{n} \sum_{j=1}^n \delta_n(p_j)$$
$$\leq 2\exp(-2) \left( \frac{1}{n-1} + \frac{1}{(n-1)^2} \right), \quad n \geq 2, \tag{38}$$

which gives

$$
\begin{aligned}
\frac{1}{n}\sum_{j=1}^{n}(1-p_j)^n &= \frac{1}{n}\sum_{j=1}^{n}\exp\left(-p_j n\right) \\
&\quad + \frac{1}{n}\sum_{j=1}^{n}\left[(1-p_j)^n - \exp\left(-p_j n\right)\right] \\
&\geq \left(\prod_{j=1}^{n}\exp\left(-p_j n\right)\right)^{\frac{1}{n}} \\
&\quad - 2\exp(-2)\left(\frac{1}{n-1} + \frac{1}{(n-1)^2}\right) \\
&= \left(\exp\left(-n\sum_{j=1}^{n}p_j\right)\right)^{\frac{1}{n}} \\
&\quad - 2\exp(-2)\left(\frac{1}{n-1} + \frac{1}{(n-1)^2}\right) \\
&= \exp\left(-1\right) \\
&\quad - 2\exp(-2)\left(\frac{1}{n-1} + \frac{1}{(n-1)^2}\right)
\end{aligned} \tag{39}
$$

for $n \geq 2$, where the AM-GM inequality is applied, and the equality holds if and only if $p_1 = p_2 = \cdots = p_n$. Hence, the right hand side of (37) $\leq 1 - \exp\left(-1\right) + 2\exp(-2)[1/(n-1) + 1/(n-1)^2]$. Since the estimate holds for any fixed Gaussian random matrix $\mathbf{W}_k$, the proof is completed. $\qquad\square$

### A.2 PERTURBATION ANALYSIS

In this section, we extend Theorem 2 to the required almost orthonormality setting, where the input sequence $\tilde{\mathbf{X}} \in \mathbb{R}^{n\times d}$ satisfies $\tilde{\mathbf{X}}\tilde{\mathbf{X}}^{\top} = \mathbf{I}_n + \mathbf{E}$, with $\mathbf{E} = [E_{ij}] \in \mathbb{R}^{n\times n}$ satisfying $|E_{ij}| \leq \epsilon \ll 1$ for any $i, j \in [n]$. We adopt the following approximation procedure:

1. Approximate the almost orthonormal input sequence with the exactly orthonormal sequence.

2. Bound the difference between attention products.

3. The desired results follow based on the stability and perturbation analysis.

(i) The first step is to approximate $\tilde{\mathbf{X}}$ with orthonormal matrices:[3]

$$
\min_{\mathbf{P}\in\mathbb{R}^{d\times n}:\,\mathbf{P}^{\top}\mathbf{P}=\mathbf{I}_n}\|\mathbf{P} - \tilde{\mathbf{X}}^{\top}\|_F, \tag{40}
$$

which can be explicitly solved in a closed form as follows.

**Lemma 5.** *Assume $d \geq n$. Let $\tilde{\mathbf{X}}^{\top} = \mathbf{U}\boldsymbol{\Sigma}\mathbf{V}^{\top}$ be the singular value decomposition (SVD) of $\tilde{\mathbf{X}}^{\top}$, where $\mathbf{U} \in \mathbb{R}^{d\times d}$ and $\mathbf{V} \in \mathbb{R}^{n\times n}$ are orthonormal and collect the singular vectors, $\boldsymbol{\Sigma} = \begin{bmatrix} \boldsymbol{\Sigma}_r & 0 \\ 0 & 0 \end{bmatrix} \in \mathbb{R}^{d\times n}$ with $\boldsymbol{\Sigma}_r = \mathrm{diag}(\sigma_1, \sigma_2, \cdots, \sigma_r)$ collecting the singular values ($\sigma_1 \geq \sigma_2 \geq \cdots \geq \sigma_r > 0$, $r = \mathrm{rank}(\tilde{\mathbf{X}}) \leq n$). Then we have*

$$
\begin{aligned}
\arg\min_{\mathbf{P}\in\mathbb{R}^{d\times n}:\,\mathbf{P}^{\top}\mathbf{P}=\mathbf{I}_n}&\|\mathbf{P} - \tilde{\mathbf{X}}^{\top}\|_F \\
&= \mathbf{U}_1\mathbf{V}^{\top},
\end{aligned} \tag{41}
$$

---

[3]This is also called the orthogonal procrustes problem (Gower & Dijksterhuis, 2004).

where $\mathbf{U}_1 := \mathbf{U}\begin{bmatrix}\mathbf{I}_n \\ 0\end{bmatrix} \in \mathbb{R}^{d\times n}$ *denotes the first $n$ columns of* $\mathbf{U}$. *Furthermore, if the input sequence* $\tilde{\mathbf{X}} \in \mathbb{R}^{n\times d}$ *is almost* orthonormal *such that* $\tilde{\mathbf{X}}\tilde{\mathbf{X}}^\top = \mathbf{I}_n + \mathbf{E}$ *with* $\mathbf{E} = [E_{ij}] \in \mathbb{R}^{n\times n}$ *satisfying* $|E_{ij}| \leq \epsilon = o(1/n^{\frac{3}{2}})$ *($\forall i,\, j \in [n]$), then* $r = \mathrm{rank}(\tilde{\mathbf{X}}) = n$, *and we have the following estimate*

$$\|\mathbf{U}_1\mathbf{V}^\top - \tilde{\mathbf{X}}^\top\|_F \leq \epsilon n^{\frac{3}{2}} = o(1). \tag{42}$$

*Proof.* First, we can derive that

$$\arg\min_{\mathbf{P}\in\mathbb{R}^{d\times n}:\,\mathbf{P}^\top\mathbf{P}=\mathbf{I}_n} \|\mathbf{P} - \tilde{\mathbf{X}}^\top\|_F^2$$

$$= \arg\min_{\mathbf{P}\in\mathbb{R}^{d\times n}:\,\mathbf{P}^\top\mathbf{P}=\mathbf{I}_n} \mathrm{trace}((\mathbf{P} - \tilde{\mathbf{X}}^\top)^\top(\mathbf{P} - \tilde{\mathbf{X}}^\top))$$

$$= \arg\min_{\mathbf{P}\in\mathbb{R}^{d\times n}:\,\mathbf{P}^\top\mathbf{P}=\mathbf{I}_n} \mathrm{trace}(\mathbf{P}^\top\mathbf{P} - \mathbf{P}^\top\tilde{\mathbf{X}}^\top$$

$$\qquad - \tilde{\mathbf{X}}\mathbf{P} + \tilde{\mathbf{X}}\tilde{\mathbf{X}}^\top)$$

$$= \arg\max_{\mathbf{P}\in\mathbb{R}^{d\times n}:\,\mathbf{P}^\top\mathbf{P}=\mathbf{I}_n} \mathrm{trace}(\tilde{\mathbf{X}}\mathbf{P})$$

$$= \arg\max_{\mathbf{P}\in\mathbb{R}^{d\times n}:\,\mathbf{P}^\top\mathbf{P}=\mathbf{I}_n} \mathrm{trace}(\mathbf{\Sigma}^\top \cdot \mathbf{U}^\top\mathbf{P}\mathbf{V}). \tag{43}$$

Let $\mathbf{S} := \mathbf{U}^\top\mathbf{P}\mathbf{V} = [S_{ij}] \in \mathbb{R}^{d\times n}$, then $\mathbf{S}^\top\mathbf{S} = \mathbf{V}^\top\mathbf{P}^\top\mathbf{U}\mathbf{U}^\top\mathbf{P}\mathbf{V} = \mathbf{I}_n$, which yields $1 = \sum_{j=1}^d S_{ji}^2 \geq S_{ii}^2$ for any $i \in [n]$. Therefore, note that

$$\mathrm{trace}(\mathbf{\Sigma}^\top \cdot \mathbf{S}) = \sum_{i=1}^r \sigma_i S_{ii} \tag{44}$$

$$\leq \sum_{i=1}^r \sigma_i |S_{ii}| \leq \sum_{i=1}^r \sigma_i, \tag{45}$$

and the equality holds when $S_{ii} = 1$ for any $i \in [r]$, we deduce that

$$\arg\max_{\mathbf{S}\in\mathbb{R}^{d\times n}:\,\mathbf{S}^\top\mathbf{S}=\mathbf{I}_n} \mathrm{trace}(\mathbf{\Sigma}^\top \cdot \mathbf{S})$$

$$= \begin{bmatrix}\mathbf{I}_n \\ 0\end{bmatrix}. \tag{46}$$

Combining with (43), we equivalently obtain

$$\arg\min_{\mathbf{P}\in\mathbb{R}^{d\times n}:\,\mathbf{P}^\top\mathbf{P}=\mathbf{I}_n} \|\mathbf{P} - \tilde{\mathbf{X}}^\top\|_F^2$$

$$= \arg\max_{\mathbf{P}\in\mathbb{R}^{d\times n}:\,\mathbf{P}^\top\mathbf{P}=\mathbf{I}_n} \mathrm{trace}(\mathbf{\Sigma}^\top \cdot \mathbf{U}^\top\mathbf{P}\mathbf{V})$$

$$= \mathbf{U}\begin{bmatrix}\mathbf{I}_n \\ 0\end{bmatrix}\mathbf{V}^\top = \mathbf{U}_1\mathbf{V}^\top, \tag{47}$$

which proves (41). To prove (42), note that $\sigma_i^2$ is the $i$-th eigenvalue of $\tilde{\mathbf{X}}\tilde{\mathbf{X}}^\top$, according to Weyl's theorem, we have

$$|\sigma_i^2 - 1| \leq \|\tilde{\mathbf{X}}\tilde{\mathbf{X}}^\top - \mathbf{I}_n\|_2 \tag{48}$$

$$= \|\mathbf{E}\|_2, \quad i \in [n]. \tag{49}$$

Since

$$\|\mathbf{E}\|_2^2 = \max_{\mathbf{z}\in\mathbb{R}^n:\,\|\mathbf{z}\|_2=1} \|\mathbf{E}\mathbf{z}\|_2^2 \tag{50}$$

$$= \max_{\mathbf{z}\in\mathbb{R}^n:\,\|\mathbf{z}\|_2=1} \sum_{i=1}^n |\mathbf{E}_{i,:} \cdot \mathbf{z}|^2 \tag{51}$$

$$\leq \max_{\mathbf{z}\in\mathbb{R}^n:\,\|\mathbf{z}\|_2=1} \sum_{i=1}^n \|\mathbf{E}_{i,:}\|_2^2 \|\mathbf{z}\|_2^2 \tag{52}$$

$$= \|\mathbf{E}\|_F^2 \leq \epsilon^2 n^2, \tag{53}$$

where $\mathbf{E}_{i,:}$ denotes the $i$-th row of $\mathbf{E}$, we get

$$|\sigma_i^2 - 1| \leq \epsilon n = o(1/\sqrt{n}), \quad i \in [n], \tag{54}$$

leading to $\sigma_i > 0$ for any $i \in [n]$, and hence $\tilde{\mathbf{X}}$ has the full rank $r = \text{rank}(\tilde{\mathbf{X}}) = n$. Therefore

$$\|\mathbf{U}_1 \mathbf{V}^\top - \tilde{\mathbf{X}}^\top\|_F^2$$

$$= \left\| \mathbf{U} \begin{bmatrix} \mathbf{I}_n \\ 0 \end{bmatrix} \mathbf{V}^\top - \mathbf{U} \boldsymbol{\Sigma} \mathbf{V}^\top \right\|_F^2$$

$$= \left\| \begin{bmatrix} \mathbf{I}_n \\ 0 \end{bmatrix} - \begin{bmatrix} \boldsymbol{\Sigma}_n \\ 0 \end{bmatrix} \right\|_F^2$$

$$= \sum_{i=1}^n |1 - \sigma_i|^2 = \sum_{i=1}^n \frac{|1 - \sigma_i^2|^2}{|1 + \sigma_i|^2} \tag{55}$$

$$\leq \sum_{i=1}^n \epsilon^2 n^2 = \epsilon^2 n^3 = o(1), \tag{56}$$

which completes the proof. $\qquad\qquad\qquad\qquad\qquad\qquad\qquad\qquad\qquad\qquad\quad\square$

(ii) As the second step, the difference between attention products can be further bounded as follows.

**Lemma 6.** *Let $\mathbf{X} := \mathbf{V}\mathbf{U}_1^\top$ with $\mathbf{V}, \mathbf{U}_1$ defined in Lemma 5. Under the same conditions in Lemma 5, and further assume $\epsilon = o(1/(n^{\frac{3}{2}}(d + d_h)))$ we have the following estimates:*

1. *For any $t > 0$, with probability at least $(1 - 2\exp(-t^2))^2$, it holds that*

$$\|\mathbf{X}\mathbf{W}_q \mathbf{W}_k^\top \mathbf{X}^\top - \tilde{\mathbf{X}}\mathbf{W}_q \mathbf{W}_k^\top \tilde{\mathbf{X}}^\top\|_2$$

$$\lesssim \epsilon n^{\frac{3}{2}}(d + d_h + t^2) = o(1). \tag{57}$$

2. $\mathbb{E}_{\mathbf{W}_k, \mathbf{W}_q} \|\mathbf{X}\mathbf{W}_q \mathbf{W}_k^\top \mathbf{X}^\top - \tilde{\mathbf{X}}\mathbf{W}_q \mathbf{W}_k^\top \tilde{\mathbf{X}}^\top\|_2 \lesssim \epsilon n^{\frac{3}{2}}(d + d_h) = o(1).$

*Here, $\lesssim$ hides positive absolute constants.*

*Proof.* Let $\mathbf{P} := \tilde{\mathbf{X}} - \mathbf{X}$. According to Lemma 5, we have $\|\mathbf{P}\|_F \leq \epsilon n^{\frac{3}{2}} = o(1)$. Then, we can derive that

$$\|\mathbf{X}\mathbf{W}_q \mathbf{W}_k^\top \mathbf{X}^\top - \tilde{\mathbf{X}}\mathbf{W}_q \mathbf{W}_k^\top \tilde{\mathbf{X}}^\top\|_2$$

$$= \|\mathbf{X}\mathbf{W}_q \mathbf{W}_k^\top \mathbf{X}^\top - (\mathbf{X} + \mathbf{P})\mathbf{W}_q \mathbf{W}_k^\top (\mathbf{X} + \mathbf{P})^\top\|_2$$

$$= \|\mathbf{P}\mathbf{W}_q \mathbf{W}_k^\top \mathbf{X}^\top + \mathbf{X}\mathbf{W}_q \mathbf{W}_k^\top \mathbf{P}^\top$$

$$\quad + \mathbf{P}\mathbf{W}_q \mathbf{W}_k^\top \mathbf{P}^\top\|_2$$

$$\leq 2\|\mathbf{P}\|_2 \|\mathbf{W}_q\|_2 \|\mathbf{W}_k\|_2 \|\mathbf{X}\|_2$$

$$\quad + \|\mathbf{P}\|_2^2 \|\mathbf{W}_q\|_2 \|\mathbf{W}_k\|_2. \tag{58}$$

Note that $\|\mathbf{P}\|_2 \leq \|\mathbf{P}\|_F \leq \epsilon n^{\frac{3}{2}} = o(1)$, $\|\mathbf{X}\|_2 = \|\mathbf{U}_1\|_2 = \|\mathbf{I}_n\|_2 = 1$, the remaining task is to estimate $\|\mathbf{W}\|_2$ for any Gaussian random matrix $\mathbf{W}$ (i.e., the entries of $\mathbf{W}$ are independent $\mathcal{N}(0,1)$ random variables). According to Theorem 4.4.5, Exercise 4.4.6 and Example 2.5.8 by Vershynin (2018), we have for any $t > 0$,

$$\|\mathbf{W}\|_2 \lesssim \sqrt{d} + \sqrt{d_h} + t, \tag{59}$$

$$\text{with probability at least } 1 - 2\exp(-t^2), \tag{60}$$

where $\lesssim$ hides positive absolute constants, and

$$\mathbb{E}\|\mathbf{W}\|_2 \lesssim \sqrt{d} + \sqrt{d_h}. \tag{61}$$

Combining with (58), we have for any $t > 0$,

$$\|\mathbf{X}\mathbf{W}_q\mathbf{W}_k^\top\mathbf{X}^\top - \tilde{\mathbf{X}}\mathbf{W}_q\mathbf{W}_k^\top\tilde{\mathbf{X}}^\top\|_2$$
$$\leq 2\|\mathbf{P}\|_2\|\mathbf{W}_q\|_2\|\mathbf{W}_k\|_2\|\mathbf{X}\|_2$$
$$+ \|\mathbf{P}\|_2^2\|\mathbf{W}_q\|_2\|\mathbf{W}_k\|_2$$
$$\lesssim (\epsilon n^{\frac{3}{2}} + \epsilon^2 n^3)(\sqrt{d} + \sqrt{d_h} + t)^2$$
$$\lesssim \epsilon n^{\frac{3}{2}}(d + d_h + t^2) = o(1), \tag{62}$$

with probability at least $(1 - 2\exp(-t^2))^2$, and

$$\mathbb{E}_{\mathbf{W}_k,\mathbf{W}_q}\|\mathbf{X}\mathbf{W}_q\mathbf{W}_k^\top\mathbf{X}^\top - \tilde{\mathbf{X}}\mathbf{W}_q\mathbf{W}_k^\top\tilde{\mathbf{X}}^\top\|_2$$
$$\leq 2\|\mathbf{P}\|_2\|\mathbf{X}\|_2 \cdot \mathbb{E}_{\mathbf{W}_q}\|\mathbf{W}_q\|_2$$
$$\cdot \mathbb{E}_{\mathbf{W}_k}\|\mathbf{W}_k\|_2 + \|\mathbf{P}\|_2^2 \cdot \mathbb{E}_{\mathbf{W}_q}\|\mathbf{W}_q\|_2$$
$$\cdot \mathbb{E}_{\mathbf{W}_k}\|\mathbf{W}_k\|_2$$
$$\lesssim (\epsilon n^{\frac{3}{2}} + \epsilon^2 n^3)(\sqrt{d} + \sqrt{d_h})^2 \tag{63}$$
$$\lesssim \epsilon n^{\frac{3}{2}}(d + d_h) = o(1), \tag{64}$$

which completes the proof. $\qquad\square$

(iii) The third step is to apply the stability and perturbation analysis.

**Proposition 1.** *(Stability of numerical ranks) Let $\sigma_{min} \neq 0$ denote the minimal non-zero singular value of a matrix $\mathbf{A}$. Then for any perturbation $\mathbf{P}$ with $\|\mathbf{P}\|_2 \leq \sigma_{min}/3$ and any $\delta \in (\sigma_{min}/3, 2\sigma_{min}/3]$, we have*

$$\mathrm{rank}(\mathbf{A}, \delta) = \mathrm{rank}(\mathbf{A} + \mathbf{P}, \delta). \tag{65}$$

*Proof.* By definition, the numerical rank $\mathrm{rank}(\mathbf{A}, \delta)$ equals to the number of singular values (of $\mathbf{A}$) no less than $\delta$. Therefore, for any $\delta \in (0, \sigma_{min}]$, $\mathrm{rank}(\mathbf{A}, \delta)$ equals to the number of non-zero singular values of $\mathbf{A}$. Let $\{\sigma_i\}$ and $\{\tilde{\sigma}_i\}$ be the singular values of $\mathbf{A}$ and $\mathbf{A} + \mathbf{P}$, respectively. According to Weyl's theorem, we have $|\sigma_i - \tilde{\sigma}_i| \leq \|\mathbf{P}\|_2 \leq \sigma_{min}/3$. Then for any $\delta \in (\sigma_{min}/3, 2\sigma_{min}/3]$, the perturbation of non-zero singular values satisfies $\tilde{\sigma}_i \geq \sigma_i - \sigma_{min}/3 \geq \sigma_{min} - \sigma_{min}/3 \geq \delta$, which is selected for counting the numerical rank, and the perturbation of zero singular values satisfies $\tilde{\sigma}_i \leq \sigma_{min}/3 < \delta$, which is not selected for counting the numerical rank. That is, $\mathrm{rank}(\mathbf{A} + \mathbf{P}, \delta)$ still equals to the number of non-zero singular values of $\mathbf{A}$, hence the desired result follows. $\qquad\square$

**Further Perturbation Analysis.** The subsequent analysis is similar, since all the remaining operations (activation, numerical rank and expectation) are *stable*. In fact, both the activation and expectation are continuous with respect to perturbations of inputs, and so does the numerical rank due to Proposition 1. Therefore, the derived upper bounds in Theorem 2 still hold for almost orthonormal input sequences.

### A.3 THE MODEL-REDUCTION EFFECT

In fact, the attention rank (the left hand side of (2)) reaches saturation when continuously increasing the head dimension $d_h$, provided an appropriate scaling (e.g. $1/\sqrt{d_h}$). Recall that the rows of $\mathbf{X}\mathbf{W}_q\mathbf{W}_k^\top\mathbf{X}^\top = \mathbf{Q}\mathbf{K}^\top$ are independent and identically distributed as $\mathcal{N}(\mathbf{0}_n, \mathbf{K}\mathbf{K}^\top)$, according to Johnson–Lindenstrauss lemma (Johnson & Lindenstrauss, 1984), we have

$$\mathbf{e}_i^\top\mathbf{K}\mathbf{K}^\top\mathbf{e}_j = \mathbf{k}_i^\top\mathbf{k}_j \tag{66}$$
$$= \mathbf{x}_i^\top\mathbf{W}_k\mathbf{W}_k^\top\mathbf{x}_j \tag{67}$$
$$\approx d_h\mathbf{x}_i^\top\mathbf{x}_j \tag{68}$$

with high probabilities when $d_h = \Omega(\log n)$, which gives

$$\mathbf{e}_i^\top\mathbf{Q}\mathbf{K}^\top/\sqrt{d_h} \sim \mathcal{N}(\mathbf{0}_n, \mathbf{K}\mathbf{K}^\top/d_h)$$
$$\approx \mathcal{N}(\mathbf{0}_n, \mathbf{X}\mathbf{X}^\top), \quad d_h = \Omega(\log n). \tag{69}$$

Table 2: The attention ranks for different data distributions: $\mathcal{N}(0,1)$, $\mathcal{N}(0,100)$, $\mathcal{U}(-1,1)$ and $\mathcal{U}(-100,100)$. Note that the normal distributions correspond with the practical NLP applications where input tokens are initially embedded with Gaussian distributions. Here, $d_h$ represents the head dimension. The "Rank / Seq Len" is the ratio of attention ranks over sequence lengths, with the standard deviation denoted by $\pm$. The "Improvement" column summarizes the successive increases in the "Rank / Seq Len" column compared to the previous row.

| $d_h$ | $\mathcal{N}(0,1)$ | | $\mathcal{N}(0,100)$ | | $\mathcal{U}(-1,1)$ | | $\mathcal{U}(-100,100)$ | |
|---|---|---|---|---|---|---|---|---|
| | Rank / Seq Len | Improvement | Rank / Seq Len | Improvement | Rank / Seq Len | Improvement | Rank / Seq Len | Improvement |
| 2 | $0.11 \pm 0.023$ | - | $0.10 \pm 0.014$ | - | $0.17 \pm 0.039$ | - | $0.09 \pm 0.016$ | - |
| 4 | $0.25 \pm 0.032$ | +0.14 | $0.23 \pm 0.029$ | +0.12 | $0.30 \pm 0.038$ | +0.13 | $0.23 \pm 0.027$ | +0.14 |
| 8 | $0.40 \pm 0.035$ | +0.15 | $0.41 \pm 0.034$ | +0.18 | $0.45 \pm 0.036$ | +0.15 | $0.38 \pm 0.028$ | +0.15 |
| 16 | $0.51 \pm 0.033$ | +0.11 | $0.52 \pm 0.036$ | +0.11 | $0.56 \pm 0.033$ | +0.11 | $0.49 \pm 0.035$ | +0.11 |
| 32 | $0.57 \pm 0.033$ | +0.06 | $0.57 \pm 0.038$ | +0.05 | $0.63 \pm 0.028$ | +0.07 | $0.56 \pm 0.031$ | +0.07 |
| 64 | $0.60 \pm 0.032$ | +0.03 | $0.61 \pm 0.032$ | +0.04 | $0.64 \pm 0.028$ | +0.01 | $0.59 \pm 0.012$ | +0.03 |
| 96 | $0.61 \pm 0.036$ | +0.01 | $0.61 \pm 0.018$ | +0.00 | $0.64 \pm 0.008$ | +0.00 | $0.60 \pm 0.050$ | +0.01 |

Due to the (positive) scaling-invariant property of $\mathrm{hardmax}$, we approximately deduce that the attention rank (the left hand side of (2)) only depends on $\mathbf{X}$ (and hence $n, d$), i.e.

$$\mathrm{rank}\left(\mathrm{hardmax}\left(\mathbf{X}\mathbf{W}_q\mathbf{W}_k^\top\mathbf{X}^\top\right)\right) \tag{70}$$

$$= \mathrm{rank}\left(\mathrm{hardmax}\left(\mathbf{Q}\mathbf{K}^\top/\sqrt{d_h}\right)\right) \tag{71}$$

$$\overset{\mathrm{d}}{\approx} \mathrm{rank}\left(\mathrm{hardmax}\left(\mathrm{rows\ of}\ \mathcal{N}(\mathbf{0}_n, \mathbf{X}\mathbf{X}^\top)\right)\right), \tag{72}$$

when $d_h = \Omega(\log n)$, where $\overset{\mathrm{d}}{\approx}$ represents the approximation in distribution. That is, increasing the head dimension beyond a certain threshold, specifically after $d_h^* = \Omega(\log n)$, results in a *limited* impact on the attention rank,

which is eventually influenced by $n$ and $d$.

This phenomenon can be understood as a manifestation of the model-reduction effect: selecting the critical configuration $d_h^* = \Omega(\log n)$ achieves optimal model efficiency, since further increasing parameters leads to *diminishing marginal utility*.

**Remark 5.** *For the constants involved in $d_h = \Omega(\log n)$, according to Johnson–Lindenstrauss lemma, it is of order $1/\epsilon^2$, where $\epsilon$ is the gap tolerance between the products of projected vectors and original vectors (i.e. the error of "$\approx$" in (66)). Additionally, there are universal constants related to $\delta$ (probability tolerance) and methods of projections. That is, for requirements of higher probabilities (smaller $\delta$), the universal constants are larger; for nonlinear projections instead of linear random projections used here, the universal constants can be potentially smaller.*

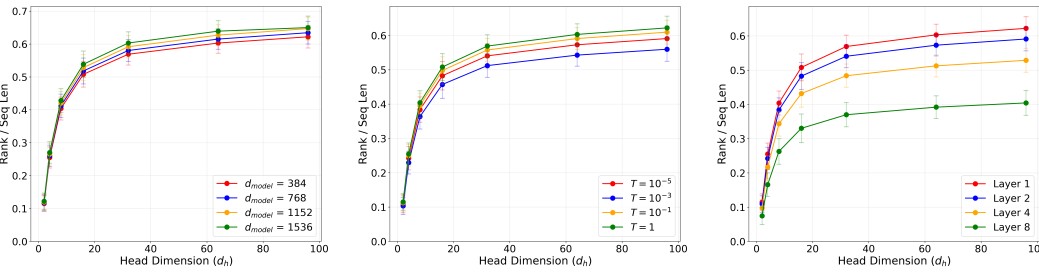

(a) The attention ranks across different model dimensions.

(b) Attention ranks across various softmax temperatures.

(c) Attention ranks across different Transformer layers.

Figure 5: Attention analysis across different configurations.

# B FURTHER DETAILS OF ABLATION STUDIES

We conduct ablation studies on both model hyper-parameters and data distributions.

## B.1 Effect of Model Dimensions

In this section, we study the effect of model dimensions on the attention rank of Transformers. We test for different dimensions $d_{\text{model}} \in \{384, 768, 1152, 1536\}$, maintaining other configurations specified in Section 2.1. The results illustrated in Figure 5a align with the phenomena observed in Figure 1, indicating a robust and consistent pattern of attention ranks across varied model dimensions.

## B.2 Effect of Softmax Temperatures

In this section, we investigate the impact of softmax temperatures on the attention rank of Transformer models. We test for different temperatures $T \in \{10^{-5}, 10^{-3}, 10^{-1}, 1\}$, and all the other configurations remain the same as those of Section 2.1.

The softmax temperature is an important factor that influences the sharpness of the attention distribution. Lower temperatures lead to more concentrated attention distributions, effectively pushing the softmax activation towards the hardmax activation. Conversely, higher temperatures yield more uniform attention distributions. Despite of these differences, our results show consistent patterns of attention ranks across all tested temperatures. This consistency, as is depicted in Figure 5b, suggests that the attention rank of Transformers is robust to variations in softmax temperatures.

## B.3 Effect of Transformers' Layers

In this section, we detail the influence of Transformers' layers on the attention rank. The experiment utilizes a model configuration with 8 layers to examine the attention rank's behavior across layers, and the other configurations are consistent with Section 2.1.

The results shown in Figure 5c exhibit a noticeable trend: with the increase of depth, the attention mechanism tends to show a more pronounced low-rank behavior. This trend is particularly evident in the deeper layers of the Transformer, suggesting that the model depth significantly influences the dynamics of attention ranks.

## B.4 Effect of Data Distributions

For a comprehensive analysis of the impact of data distributions on the attention rank of Transformers, we numerically study a range of data distributions including normal distributions $\mathcal{N}(0, 1)$ and $\mathcal{N}(0, 100)$, as well as uniform distributions $\mathcal{U}(-1, 1)$ and $\mathcal{U}(-100, 100)$. These distributions are selected to mimic common scenarios in NLP applications, where input tokens are typically embedded using Gaussian distributions. The model configurations used in these experiments are consistent with Section 2.1.

Our findings reveal the remarkable robustness of the attention rank with respect to data distributions, as is evidenced by consistent patterns of attention ranks across all tested data distributions in Table 2. It is particularly notable for the normal distributions ($\mathcal{N}(0, 1)$ and $\mathcal{N}(0, 100)$), which show similar patterns of attention ranks and imply that the initial Gaussian embeddings of input tokens do not significantly influence the attention mechanism's efficacy. The uniform distributions $\mathcal{U}(-1, 1)$ and $\mathcal{U}(-100, 100)$ follow the same trend, reinforcing the model's insensitivity to the nature of data distributions. These results underscore the robustness of Transformer models to variations in data distributions, which is a crucial factor for real-world applications.

## B.5 Numerical Verifications on the Orthonormality

To validate the orthonormality assumption used in our theoretical analysis, we conduct numerical experiments to measure the orthogonality of input sequences across different datasets and dimensions.

We use the mean Frobenius norm as the orthogonality measure for tensors with various dimensions. Specifically, we compute $\frac{1}{n^2}\|Q - I\|_F$, where $n$ is the sequence length, $Q$ denotes the cosine similarity matrix between input tokens, and $I$ is the identity matrix. Lower mean Frobenius norms indicate that the tokens in the tensor are more orthonormal, which aligns with our theoretical assumptions.

The experiments are conducted on both synthetic Gaussian random data and real-world datasets including CIFAR-10 and CIFAR-100 (after passing through an initialized embedding layer). As shown in Figure 6, both Gaussian random data and the real-world datasets exhibit relatively small mean Frobenius norms across different head dimensions $d_h$. This observation confirms that the input sequences are indeed nearly orthonormal in practice, validating the orthonormality assumption underlying our theoretical analysis. These results demonstrate that the almost orthonormal condition is not merely a theoretical convenience but reflects actual properties of embedded data in Transformer models, thereby supporting the practical relevance of our theoretical findings.

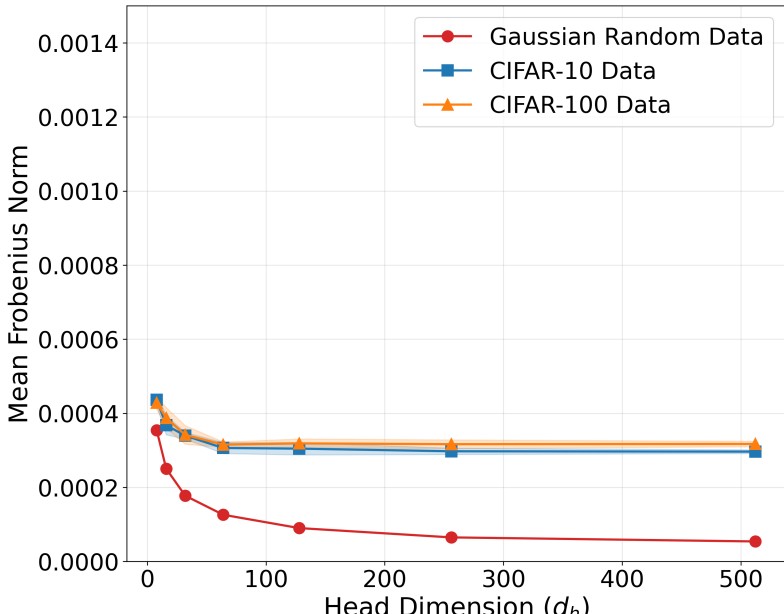

Figure 6: Orthogonality measure across different dimensions for Gaussian random, CIFAR-10, and CIFAR-100 data.

## C  FURTHER DETAILS ON REAL-WORLD EXPERIMENTS

### C.1  DETAILED EXPERIMENTAL SETUP

For the computer vision (CV) experiments, we set the feed-forward hidden dimension as 384. The model depth is 7. For the learning, the batch sizes are 128 for training and 1024 for evaluation. The initial learning rate is set as $10^{-3}$. We perform the train-validation-test split on the datasets following official guidelines. To align with real-world applications, various techniques are integrated, including label smoothing and auto-augmentation. Moreover, the experiments also involve advanced regularization methods (specifically, CutMix (Yun et al., 2019) and MixUp (Zhang et al., 2018)) to enhance the models' generalization performance. We conduct all experiments on a single machine with the NVIDIA GeForce RTX 3090 (24 GB).

### C.2  MODEL-REDUCTION: FIXED MODEL DIMENSIONS

In this section, we present a detailed set of experimental results on the performance of Vision Transformers (ViTs) with fixed model dimensions on the CIFAR-10, CIFAR-100 and SVHN dataset to elucidate the model-reduction effect on various datasets. We present these experimental results in Figure 7, Figure 8, and Figure 9. These results further corroborate and align with the findings discussed in the main text, demonstrating the existence of saturation in model performance when fixed model dimensions.

Table 3: The final accuracy for different models on varied datasets.

| Configurations | | Final accuracy | | | | |
|---|---|---|---|---|---|---|
| Datasets | $d_{\text{model}}$ | Head = 1 | Head = 2 | Head = 4 | Head = 8 | Head = 16 |
| Cifar-10 | 192 | 0.8836 | 0.8981 | 0.9004 | 0.9013 | 0.8932 |
| Cifar-10 | 384 | 0.8795 | 0.8924 | 0.8977 | 0.9000 | 0.8997 |
| Cifar-100 | 192 | 0.6316 | 0.6435 | 0.6454 | 0.6470 | 0.6378 |
| Cifar-100 | 384 | 0.6280 | 0.6497 | 0.6685 | 0.6680 | 0.6671 |
| SVHN | 192 | 0.9684 | 0.9717 | 0.9737 | 0.9739 | 0.9724 |
| SVHN | 384 | 0.9721 | 0.9723 | 0.9713 | 0.9730 | 0.9757 |

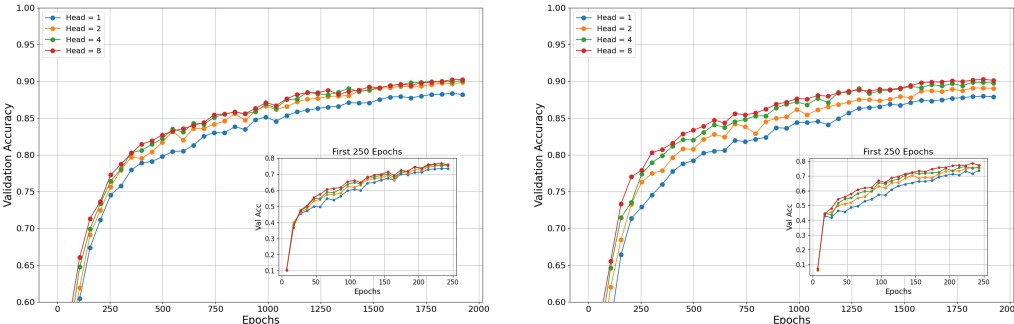

Figure 7: The validation accuracy of ViTs on the CIFAR-10 dataset with the model dimensions 192 (left) and 384 (right).

**Final Accuracy.** We also summarize the final accuracy achieved by each experiment in Table 3. These results indicate that with the constraint $d = d_{\text{model}} = h \times d_h$, a smaller number of heads $h$ results in a larger head dimension $d_h$, potentially exceeding the critical head dimension to achieve the rank saturation for each head. Namely, most of the heads may have reached the saturation point, leading to the redundancy in modeling parameters. On the contrary, as the number of heads increases, the Transformer model with reduced head dimensions gradually avoids rank saturation (and potential parameter redundancy), leading to more portions of "effective" ranks for modeling, which yields improved experimental results.

## C.3 MODEL-REDUCTION: FIXED NUMBER OF HEADS

In this section, we present supplementary results on the performance of Vision Transformers (ViTs) in varied model dimensions (with a fixed number of heads) on the CIFAR-10, CIFAR-100 and SVHN dataset to elucidate the model-reduction effect on various datasets. We present these experimental results in Figure 11, Figure 12, and Figure 13. Notably, although the initial improvement in the validation accuracy is pronounced as the head dimension $d_h$ increases within relatively small values, this improvement plateaus for appropriately large values of $d_h$, indicating diminishing returns with further increments in modeling parameters. These observations align with our theoretical justifications on the model-reduction effect, suggesting an optimal range for head dimensions that balance the model performance with parameter efficiency.

**Relation to Attention Ranks.** The experiments focus on evaluating the model-reduction effect on the CIFAR-10 dataset with a fixed number of heads $h = 8$ and varying head dimensions $d_h$. We test 5 different values of $d_h$: $d_h = 2, 4, 8, 16, 32$.

In Figure 10a, it is shown that while validation accuracy improves significantly as $d_h$ increases within relatively small values, this improvement plateaus for appropriately large values of $d_h$, showcasing diminishing returns with further increments in modeling parameters. The optimal configuration occurs at $d_h^* = 16$, as $d_h = 32$ yields marginal improvements in accuracies.

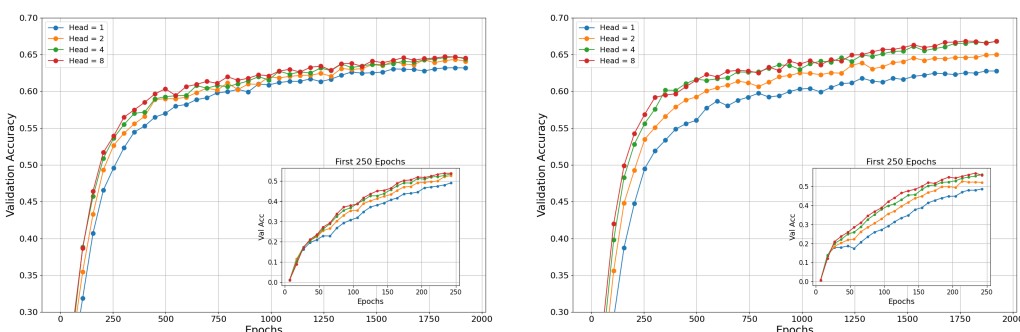

Figure 8: The validation accuracy of ViTs on the CIFAR-100 dataset with the model dimensions 192 (left) and 384 (right).

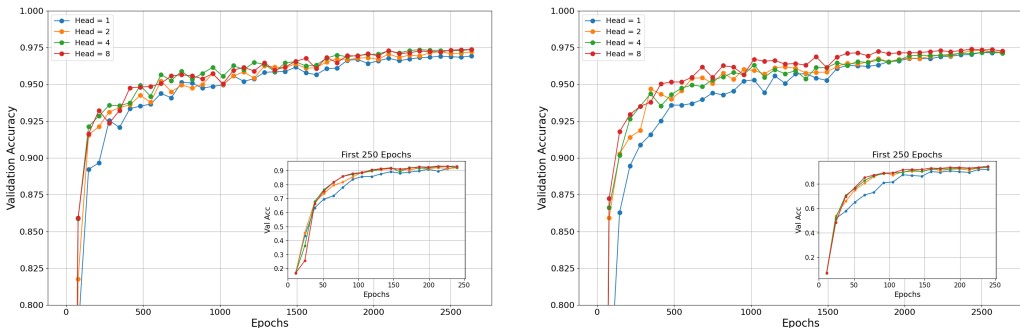

Figure 9: The validation accuracy of ViTs on the SVHN dataset with the model dimensions 192 (left) and 384 (right).

Notably, the corresponding attention ranks[4] in Figure 10b also exhibit saturation when $d_h \geq d_h^* = 16$, which aligns with the performance trend observed in Figure 10a. We observe that smaller values of $d_h$ lead to significant improvements in attention ranks as $d_h$ increases. However, when the values of $d_h$ become larger ($d_h \geq 16$), further increases have marginal effects on attention ranks. This correlation between attention rank saturation and performance plateauing validates our theoretical analysis of the model-reduction effect. In other words, once the attention rank reaches saturation, further increasing $d_h$ has limited impact on the final model performance, and hence leads to the model redundancy.

## D    THE USE OF LARGE LANGUAGE MODELS

The human authors prepared the original drafts. Subsequently, large language models were employed to refine the text, improving linguistic quality, structural coherence, and overall clarity. After the model's adjustments, the authors performed a comprehensive final review and confirming that the manuscript accurately represented our methods and results.

---

[4]The attention ranks are calculated for the first-layer attention matrices on a mini-batch of CIFAR-10 images for different head dimensions, averaged over all heads and multiple varied random seeds.

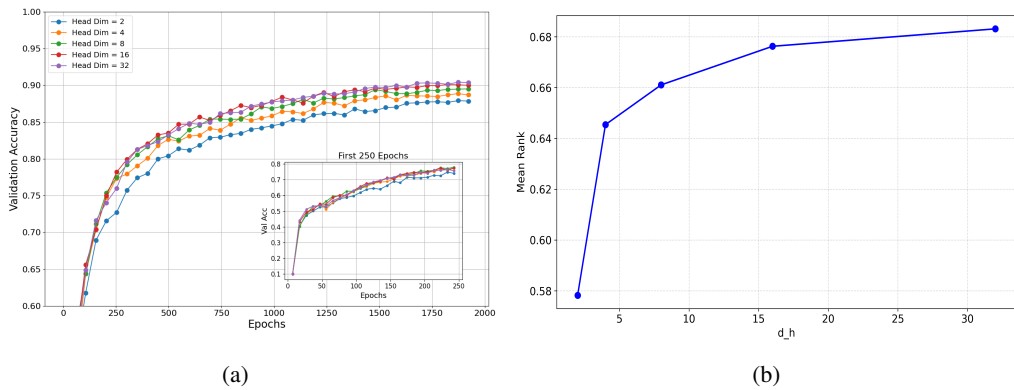

Figure 10: Real-world experiments on CIFAR-10 with fixed number of attention heads and varying head dimensions. (a) model accuracy as a function of head dimension. (b) attention rank evolution with increasing head dimension. The correlation between attention ranks and model performance is clearly demonstrated.

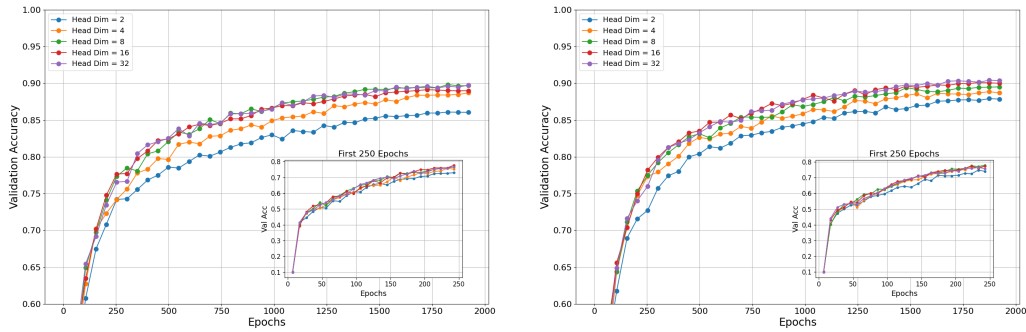

Figure 11: The validation accuracy of ViTs on the CIFAR-10 dataset with 4 heads (left) and 8 heads (right).

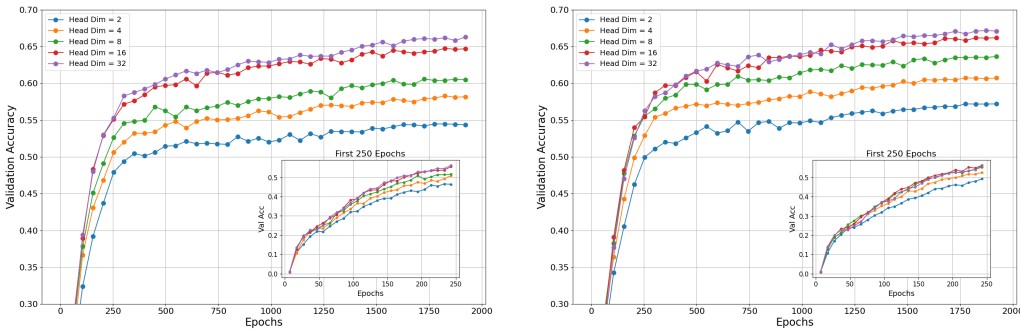

Figure 12: The validation accuracy of ViTs on the CIFAR-100 dataset with 4 heads (left) and 8 heads (right).

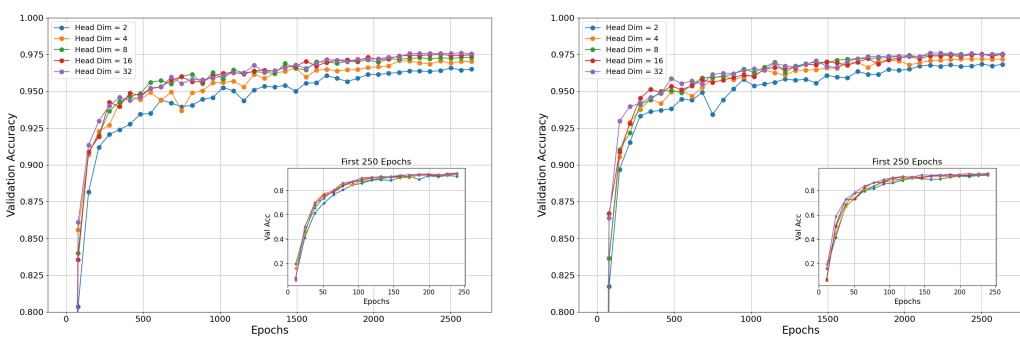

Figure 13: The validation accuracy of ViTs on the SVHN dataset with 4 heads (left) and 8 heads (right).

