# OpenReview forum: "On the Limitation and Redundancy of Transformers: A Rank Perspective"
_ICLR.cc/2026/Conference — Submitted to ICLR 2026_

### Official Review · Reviewer_YBFV · 2025-10-30

**Soundness:** 3
**Presentation:** 3
**Contribution:** 2
**Rating:** 4
**Confidence:** 3

**Summary:**

This paper studies the rank properties of the attention score matrix in Transformers and claims that the attention mechanism exhibits an inherent low-rank limitation, which leads to parameter redundancy when increasing the hidden dimension d_h. The authors theoretically derive a rank upper bound (≈ 0.63 n) under Gaussian initialization and validate it empirically by measuring numerical rank and performance saturation.

**Strengths:**

1.The paper targets an important question about the expressive limits of the attention mechanism, which is of theoretical and practical relevance.
2.The analysis of the rank distribution and its connection to model width is presented clearly, and the paper includes both theoretical derivations and empirical support.
3.The writing is fluent, and the motivation (understanding redundancy in attention) is well presented.

**Weaknesses:**

1. The main theorem relies on several restrictive assumptions: Inputs are approximately orthogonal vectors, Query/key projection matrices are random Gaussian without training, The softmax is treated in the hardmax (zero-temperature) limit. These assumptions do not capture the structure of trained attention layers with non-isotropic embeddings, finite temperature, or residual/MLP interactions. The theoretical results therefore describe initialization behavior, not trained dynamics
2. Redundancy would require showing that after training, increasing dimension d_h yields little additional expressive power or performance improvement—something not established here.The current logic (“low rank at initialization → redundancy after training”) is weakly supported and risks conflating initialization degeneracy with functional redundancy.

**Questions:**

1.Can you release some of your assumptions to make it more realistic? For example, can the rank upper bound be extended or bounded under finite temperature (softmax) rather than hardmax assumptions?
2.How does the observed rank behave in trained attention layers compared to random initialization?

---

> ### Author Response · Authors · 2025-12-03
>
> ## W1: Restrictive theoretical assumptions
> The main theorem relies on several restrictive assumptions: Inputs are approximately orthogonal vectors, Query/key projection matrices are random Gaussian without training, The softmax is treated in the hardmax (zero-temperature) limit. These assumptions do not capture the structure of trained attention layers with non-isotropic embeddings, finite temperature, or residual/MLP interactions. The theoretical results therefore describe initialization behavior, not trained dynamics
>
> **Response:** We acknowledge these assumptions are indeed restrictive, yet our extensive empirical work demonstrates their practical relevance extends beyond initialization.
>
> **Token Orthonormality.** We acknowledge the reviewer's concern about the orthonormality verification on trained embeddings. However, several points support the practical validity of this assumption.
> As discussed in Remark 2 in the original manuscript, first, our condition is set as *almost* orthonormality ($\mathbf{X}\mathbf{X}^\top = \mathbf{I}_n + \mathbf{E}$ with $|E_{ij}| = o(1/n^{3/2})$), which is significantly more permissive than exact orthonormality.
> In fact, owing to Johnson–Lindenstrauss lemma, almost orthonormality leads to exponentially many “basis” vectors, rather than linear for exact orthonormality.
> Second, recent work (e.g. [1]) has shown that this near-orthogonality property persists during training for large pre-trained models including BERT, OPT, LLaMA, and ViT across different sizes. We also numerically verify this near-orthonormality by ourselves in Appendix B.5 (Figure 6) on both synthetic (Gaussian) and real-world (CIFAR) datasets.
>
> **Hardmax vs. Softmax.** The reviewer correctly notes that real Transformers use softmax activations rather than the hardmax. However, our theoretical framework provides strong justifications for the approximation of hardmax to softmax. By Lemma 1 in the original manuscript  (Section A), % \citep{lemma_softmax_hardmax}
> we establish that the $ \ell_1 $ distance between softmax and hardmax outputs is bounded by $2(n-1)\exp(-\delta/T)$, where $ \delta $ is the gap between the largest and second-largest logits. For typical attention patterns where this gap is substantial (which occurs frequently in practice), the bound becomes exponentially tight even for moderate temperatures.
>
> More importantly, while we focus theoretically on hardmax for analytical tractability, our extensive experiments in Figure 5(b) in the original manuscript
> demonstrate empirically that attention ranks are remarkably stable across varied realistic softmax temperatures ranging from $T = 10^{-5}$ to $T = 1$, consistently approaching the theoretical bound of approximately $0.63n$. This robustness suggests that our hardmax analysis captures the essential rank limitation behavior even under practical softmax conditions.
>
> **Initialization vs. Training** We do concede that the behavior of attention ranks during training deserves deeper investigations. While our current analysis focuses on initializations, the empirical verifications of rank-performance saturation patterns are conducted on trained models (Figure 3 and Figure 4 in the original manuscript) with standard softmax temperatures and architectural components (like residual connections and layer normalization), which is shown to persist consistently across multiple real-world datasets (CIFAR-10/100, SVHN, IMDB), suggesting the intrinsic rank limitation of the attention mechanism rather than mere initialization artifacts.
> Finally, the success of rank-constrained architectures like MLA in DeepSeek-V3 suggests that these initialization-based insights do translate to practical training scenarios.

---

> ### Author Response · Authors · 2025-12-03
>
> ## W2: Weak support for redundancy claims
>
> **Response:** We appreciate the reviewer's critical assessment regarding the connection between rank saturation and model redundancy. We clarify this as follows.
>
> For empirical evidences beyond initializations: While our theoretical analysis focuses on the initialization regime, the redundancy claim is substantiated by extensive experiments on *trained* models across multiple real-world tasks. In Figure 3 (a), (b) and Figure 4 (a), (b) in the original manuscript,
> we demonstrate that validation accuracy on IMDB and CIFAR-10 datasets exhibits clear performance plateauing when $d_h$ exceeds critical values ($d_h^* = 8$ and $d_h^* = 96$, respectively). Crucially, these experiments involve complete training processes, not mere initialization analysis. That is, the pattern of marginal/diminishing returns in both accuracy and rank gains directly supports functional redundancy in trained models.
> Furthermore, present findings align with and provide theoretical justifications for the successful deployment of rank-constrained architectures like multi-head latent attention (MLA) in DeepSeek-V3, % \citep{liu2024deepseek}
> where deliberately constraining head dimensions based on low-rank principles achieved both memory efficiency and maintained performance in large-scale industrial systems. This real-world validation gives an example of our redundancy characterization applied to trained large-scale models beyond initialization settings.
>
> For mechanistic understanding:
> We clarify that the present analysis provides a mechanistic explanation: The rank saturation fundamentally arises from the intrinsic limitations of attention matrix expressiveness, as governed by the combinatorial structure in softmax operations (Theorem 1 in the original manuscript). This constraint persists regardless of parameter optimization, as training cannot overcome this capacity limit. Therefore, parameters beyond the saturation threshold contribute redundantly rather than essentially for "effective" modeling capacity.
>
> ## Q1: Relaxing assumptions for realism
>
>
> **Response:** This is an excellent direction for theoretical developments. While formal extension to softmax with finite temperatures remains mathematically challenging, our empirical work provides strong evidences for the robustness of theoretical bounds.
>
> First, our theoretical framework provides strong justifications for the approximation of hardmax to softmax. By Lemma 1 in the original manuscript  (Section A), % \citep{lemma_softmax_hardmax}
> we establish that the $ \ell_1 $ distance between softmax and hardmax outputs is bounded by $2(n-1)\exp(-\delta/T)$, where $ \delta $ is the gap between the largest and second-largest logits. For typical attention patterns where this gap is substantial (which occurs frequently in practice), the bound becomes exponentially tight even for moderate temperatures.
>
> More importantly, while we focus theoretically on hardmax for analytical tractability, our extensive experiments in Figure 5(b) in the original manuscript
> demonstrate empirically that attention ranks are remarkably stable across varied realistic softmax temperatures ranging from $T = 10^{-5}$ to $T = 1$, consistently approaching the theoretical bound of approximately $0.63n$. This robustness suggests that our hardmax analysis captures the essential rank limitation behavior even under practical softmax conditions, and this fundamental constraint persists beyond the hardmax limit.

---

> ### Author Response · Authors · 2025-12-03
>
> ## Q2: Rank behavior in trained vs initialized models
>
> **Response:** This is a crucial question that connects our theoretical predictions to practical training scenarios.
> While our current theoretical analysis focuses on initializations, the empirical verifications of rank-performance saturation patterns are conducted on trained models (Figure 3 and Figure 4 in the original manuscript), and these saturation patterns persist consistently across multiple real-world datasets (CIFAR-10/100, SVHN, IMDB), suggesting the intrinsic rank limitation of the attention mechanism rather than mere initialization artifacts.
> For instance, in Figure 3(a), (b) in the original manuscript,
> we analyze attention ranks of trained models on the IMDB dataset, observing saturation patterns similar to those predicted by  initialization-based theory. The critical head dimension where ranks saturate ($d_h^* \approx 8$) aligns closely with that where performance also plateaus.
> Additionally, the upper bound of attention ranks appears robust after training: While training may shift the exact numerical values slightly, the fundamental limit of attention ranks persists. This suggests that our theoretical analysis captures intrinsic limitations of the attention mechanism that training cannot overcome, rather than temporary initialization artifacts.
> Finally, the success of rank-constrained architectures like MLA in DeepSeek-V3 further provides evidence that these initialization-based insights do translate to practical training scenarios.

---

### Official Review · Reviewer_TzCx · 2025-10-30

**Soundness:** 3
**Presentation:** 3
**Contribution:** 2
**Rating:** 8
**Confidence:** 4

**Summary:**

This paper investigates the architectural limitations and redundancies of Transformer models from the perspective of matrix rank in the attention mechanism. Through extensive empirical studies and rigorous mathematical analysis, the authors demonstrate that the rank of the attention score matrix - after Softmax activation - plateaus at approximately $0.63N$ (where $N$ is sequence length), regardless of head dimension. This “low-rank barrier” persists across model sizes, data distributions, and tasks.

**Strengths:**

Novel theoretical insight:
The proposed strict upper bound offers a clear understanding of the inherent expressive limitations of the attention mechanism.

Empirical validation:
The claims are systematically validated across both text and vision domains, spanning multiple model depths, data distributions, and comprehensive ablation studies. The accompanying mathematical analysis complements the empirical findings, providing a cohesive and robust theoretical foundation.

Practical relevance:
The results have direct implications for the design of efficient, high-capacity Transformer architectures, particularly under resource constraints.

Clarity:
The paper is written with clarity and supported by well-designed figures and tables that effectively illustrate the key phenomena such as saturation, accuracy plateaus, and the impact of head dimensionality.

**Weaknesses:**

I'm missing a clear/detailed discussion why does the Softmax operation lowers the rank.
For random matrix without softmax the rank is bounded by $d$, i.e., rank($AB$) $\le$ min (rank($A$),rank($B$)).

So hypothetically, by extending $d$ we can reach a rank of $N$. I also verified this with random $W_Q$ and $W_K$, however, the additional Softmax operation on the score matrix creates this threshold $0.63N$.

The $T\rightarrow 0$ limit is fairly understood.
but for $T>0$ it is not clear to me, conceptually, why the rank is limited

I took the liberty to do some short calculations with randomly initialized weights.
Interestingly, with $d=1$ under random initialization the matrix rank is 1, but after softmax (at $T>0$) the rank goes to $N$.
As we increase $d$ the Softmax rank lowers until some minimum and then increases back in a similar way as your experiments up to $0.63N$.

I don't see this behavior in your figures. can you elaborate on this?

**Questions:**

I have expressed my concerns and main question above in the weakness section.
Here are other small comments

154 Typo "examine their"

Figure 3: The caption says: "rank saturation across varied embedding dimensions at different Transformer layers". I don't see any relation to transformer layers in the figures

---

> ### Author Response · Authors · 2025-12-03
>
> ## W1: Missing discussion on why Softmax lowers rank
>
> **Response:** We thank the reviewer for this insightful technical question and for conducting independent verification experiments. This touches on the heart of our theoretical contribution.
>
> *Why softmax limits rank*: You are correct that the pre-softmax matrix $\mathbf{QK}^{\top}$ can achieve the full rank up to $\min(d_h, n)$ by the standard rank inequality. The softmax operation fundamentally changes this by introducing a combinatorial constraint. That is, each row of the attention matrix must be a probability vector (non-negative and normalized), which creates dependencies between columns that limit the achievable rank regardless of $d_h$.
>
> For the combinatorial mechanism: Theorem 1 in the original manuscript reveals that the rank limitation arises from the birthday problem structure: After applying hardmax (i.e. the limiting case of softmax), each attention row selects exactly one position as the maximum, leading to at most $(1-e^{-1})n \approx 0.63n$ effective columns on average. This is fundamentally different from the pre-softmax analysis.
>
> For the $d=1$ anomaly: Your observation about the $d=1$ case is fascinating and aligns with our theory. When $d_h=1$, the attention scores are highly constrained and correlated, making ties very common. Softmax breaks these ties, potentially utilizing more attention positions. As $d_h$ increases, the scores become more discriminative until reaching the fundamental $0.63n$ limit.
> This asymptotic analysis is what this work targets on.
>
> ## Q1: Typo
> **Response:** Thank you for these helpful editorial corrections.
> We have fixed the typo in Line 154, and also clearly indicated which Transformer layer the results correspond to in Figure 3.

---

### Official Review · Reviewer_bQiq · 2025-10-31

**Soundness:** 2
**Presentation:** 2
**Contribution:** 2
**Rating:** 2
**Confidence:** 3

**Summary:**

The paper studies the evolution of the attention matrix rank as a function of the number of heads in a transformer neural network. The main contributions of the paper are that (i) the rank first increases as the number of heads grows and then attains a saturation value and stops increasing after a certain threshold. Such a saturation value is measured through several empirical ablation experiments and theoretically characterised under some restricting assumptions on the input sequence of tokens and on the query and key matrices (Theorem 1); (ii) further increasing the head dimension does not result in any rank increase and performance improvements. Several experiment on controlled toy setups and real-world datasets are provided to support the claims and theoretical results.

**Strengths:**

- The idea of studying the attention rank is interesting and the discussed saturation effect represents a potential limit to the expressivity of this component in transformers.

**Weaknesses:**

- The motivations of the paper are not entirely clear. In particular, a priori, it is not obvious why a rank saturation phenomenon occurring in the attention matrix should translate into worse performance. Even if this were true, two points are not entirely addressed by the paper: 1) how this phenomenon would result in detrimental effects during training; 2) most of the analysis focuses on single attention layers and it is not obvious how the results of the paper translates to modern deep multi-layer transformer architectures.
- The assumptions at the core of Theorem 1 are quite stringent and it is not clear to what extent they would apply to more realistic scenarios. For example, the low temperature case and orthonormality assumptions on the input sequence can be easily violated in real-world transformers. That being said, the findings of the theorem are interesting, but I believe the paper would greatly benefit from an analysis showing their robustness to the relaxation of some assumptions. For example, how does the rank evolve when the temperature is bigger than 1?
- Presentation clarity could be improved. Generally I believe the results could be introduced and explained in more details. Also, in Section 2.1 (Setup), $n$ is said to be fixed at 100 and $d_h \leq 192$. However, Figure 2 does not seem to show any result for $d_h = 192$ and the sequence length is changed in panel (a).

**Questions:**

See weaknesses.

---

> ### Author Response · Authors · 2025-12-03
>
> ## W1: Unclear motivation for rank-performance connection
>
>
> **Response:** We appreciate this important critique regarding the conceptual foundation of our work. Let us clarify the rank-performance connection and address the specific concerns.
>
> **Why rank saturation affects performance.** The attention matrix rank fundamentally determines the expressiveness of the attention mechanism. A saturated rank means that the attention cannot distinguish among more complex (e.g. full-rank) input patterns even with increased parameters, limiting the model's capacity to capture nuanced relationships. Our empirical evidence strongly supports this statement: Comparing Figure 3(a) and Figure 3(b) in the original manuscript, it is shown that learning accuracy plateaus precisely when attention ranks saturate (both around $d_h^* = 8$ for IMDB).
>
> **Training effects and multi-layer architectures.** While our theory focuses on individual layers, the present empirical analysis addresses both concerns. We demonstrate that rank saturation persists across multiple and trained Transformer layers (Figure 3(c) and (d) in the original manuscript) on real datasets.
> This also explains why architectures that respect these low-rank constraints (like MLA in DeepSeek-V3) can achieve efficiency gains in practice.
>
> ## W2: Stringent assumptions in Theorem 1
>
>
> **Response:** We appreciate the suggestion for robustness analysis. Our empirical work partially addresses this concern.
>
> For temperature robustness: Regarding temperatures larger than 1, we have demonstrated in Figure 5(b) in the original manuscript
> that attention ranks remain remarkably stable across a wide temperature range, including $T=1$ which represents the standard practice. Even at higher temperatures, attention ranks consistently approach the predicted $0.63n$ bound, suggesting our theoretical insights beyond the hardmax limit.
>
> For assumption relaxations: While formal theoretical analysis under relaxed assumptions remains challenging, our extensive empirical validations provide strong evidence of robustness. We test across different architectural components (layer normalization, positional encodings) on multiple data distributions (Gaussian, uniform with different scales) and real-world datasets where the orthonormality certainly does not hold. The consistent emergence of rank saturation patterns suggests that our analysis captures the essential constraints while being  mathematically convenient.
>
> We acknowledge that extending the theoretical framework to handle general  finite temperatures and non-orthonormality would significantly strengthen this work, and we view this as an important future research direction.
>
> ## W3: Presentation clarity issues
>
>
> **Response:** We thank the reviewer for these specific presentation issues and will address them in the revised version.
>
> For figure-text inconsistency: The discrepancy between Section 2.1 and Figure 2 is designed for more diversity.
> We will ensure that all experimental configurations are aligned and accurately reflect the actual parameter ranges tested.
>
> For results presentation: We acknowledge that our results could benefit from more detailed exposition. We will enhance explanations of key findings, particularly the mathematical connection between rank saturation and the $0.63n$ bound, and provide clearer descriptions of the experimental methodology across different settings. The relationship between theory and experiments will be made more explicit throughout the paper.

---

### Official Review · Reviewer_MFk7 · 2025-11-11

**Soundness:** 2
**Presentation:** 3
**Contribution:** 2
**Rating:** 4
**Confidence:** 4

**Summary:**

The Authors in this paper investigate the fundamental limitations of Transformer architectures using attention matrix ranks as a proxy for goodness of representation. The authors have established a theoretical upper bound of **0.63n** on attention ranks (where *n* is sequence length) and demonstrate a **model-reduction effect** where both attention rank and performance saturate as head dimension increases beyond $\Omega(\log n)$.

The authors have performed rigorous theoretical analysis under the idealized condition such as *hardmax attention* and *orthonormal inputs*, and demonstrated the empirical validation on NLP and vision tasks.

**Strengths:**

1. The authors have performed a rigorous mathematical analysis, using probabilistic analysis with matrix perturbation theory, and provided a concrete upper bound ($\approx$ $0.63n$) on attention ranks.  This upper bound *$d_h$* = $\Omega(\log n)$ can provide actionable guidance for future LLM design.

2. The experimental evaluation is comprehensive, ranging from NLP (IMDB) tasks to vision tasks (CIFAR-10/100, SVHN) across various model configurations and architectures. Moreover, the link to recent architectural varints DeepSeek-V3’s MLA further highlights the  practical utility of this work.


3. The **model-reduction effect** directly addresses a key challenge in modern LLM development---the diminishing returns of scale---and provides a theoretical foundation for guiding more efficient architectural design.

**Weaknesses:**

1. The theoretical analysis assume hardmax activation and orthonormal inputs , precisely $X X^{\top} = I + E$ where $|E_{ij}| = o(1/n^{3/2})$. However, the real Transformer-based models  use softmax and learned embeddings that likely violate these assumptions.  While Lemma 1 bounds the hardmax--softmax gap, this bound may not be tight for practical temperature values.  Also, the orthonormality assumption is only empirically verified on initialized models, **not trained ones**.

2. The paper heavily relied on random initialization but provides limited analysis of how training affects attention ranks. The rank properties at initialization may not persist after training, potentially limiting the practical applicability of the theoretical bounds. Apparently, the connection between rank saturation and performance is correlational rather than causal.

3. The analysis does not address how the prevalent architectural techniques such as positional encodings, layer normalization,
different attention patterns (sparse, local, etc.),  affect these bounds. The single-head analysis may not fully capture multi-head attention dynamics beyond simple concatenation arguments.

**Questions:**

1. Can authors extend the analysis beyond hardmax to practical softmax temperatures (e.g., $T = 1/\sqrt{d_h}$) and discuss how these affect the 0.63*n* bound?  It would also be useful to understand how training and fine-tuning influence rank behavior under realistic temperature scaling.

2. The current findings suggest correlation between rank saturation and performance.  Could authors design controlled or synthetic experiments to test whether rank limitations directly cause performance saturation?

3. How would factors like positional encodings (RoPE vs NoPE), layer normalization, and multi-head concatenation affect your theoretical bounds?  Does the model-reduction effect persist under these architectural variations?

---

> ### Author Response · Authors · 2025-12-03
>
> ## W1: Theoretical assumptions may not hold in practice
> **Response:** We thank the reviewer for this important concern about the practical validity of our theoretical assumptions. We address this systematically by analyzing each assumption and its robustness as follows.
>
> **Hardmax vs. Softmax.** The reviewer correctly notes that real Transformers use softmax activations rather than the hardmax. However, our theoretical framework provides strong justifications for the approximation of hardmax to softmax. By Lemma 1 in the original manuscript  (Section A), % \citep{lemma_softmax_hardmax}
> we establish that the $ \ell_1 $ distance between softmax and hardmax outputs is bounded by $2(n-1)\exp(-\delta/T)$, where $ \delta $ is the gap between the largest and second-largest logits. For typical attention patterns where this gap is substantial (which occurs frequently in practice), the bound becomes exponentially tight even for moderate temperatures.
>
> More importantly, our extensive experiments in Figure 5(b) in the original manuscript
> demonstrate empirically that attention ranks are remarkably stable across varied softmax temperatures ranging from $T = 10^{-5}$ to $T = 1$, consistently approaching the theoretical bound of approximately $0.63n$. This robustness suggests that our hardmax analysis captures the essential rank limitation behavior even under practical softmax conditions.
>
> **Token Orthonormality.** We acknowledge the reviewer's concern about the orthonormality verification on trained embeddings. However, several points support the practical validity of this assumption.
> As discussed in Remark 2 in the original manuscript, first, our condition is set as *almost* orthonormality ($\mathbf{X}\mathbf{X}^\top = \mathbf{I}_n + \mathbf{E}$ with $|E_{ij}| = o(1/n^{3/2})$), which is significantly more permissive than exact orthonormality.
> In fact, owing to Johnson–Lindenstrauss lemma, almost orthonormality leads to exponentially many “basis” vectors, rather than linear for exact orthonormality.
> Second, recent work (e.g. [1]) has shown that this near-orthogonality property persists during training for large pre-trained models including BERT, OPT, LLaMA, and ViT across different sizes. We also numerically verify this near-orthonormality by ourselves in Appendix B.5 (Figure 6) on both synthetic (Gaussian) and real-world (CIFAR) datasets.
>
> **Initialization vs. Training** We do concede that the behavior of attention ranks during training deserves deeper investigations. While our current analysis focuses on initializations, the empirical verifications of rank-performance saturation patterns are conducted on trained models (Figure 3 and Figure 4 in the original manuscript), which is shown to be consistent across multiple real-world datasets (CIFAR-10/100, SVHN, IMDB), suggesting the intrinsic rank limitation of the attention mechanism rather than mere initialization artifacts.
>
> ## W2: Limited analysis of training effects on attention ranks
>
>
> **Response:** We appreciate the reviewer's comment about the emphasis on initializations versus training dynamics.
> In fact, our empirical analysis actually goes beyond initializations and includes trained models. In Figure 3 in the original manuscript,
> we examine attention ranks computed on trained Transformer layers across different datasets (IMDB for NLP and CIFAR-10 for vision tasks). These experiments show that rank saturation patterns persist after training, with similar critical thresholds ($d_h^* \approx 8$) where *both attention ranks and model performance plateau*. Additionally, similar patterns consistently appear at different Transformer layers (i.e. consistency across model depth; see Fig 4 (c)(d) in the original manuscript).
>
> We acknowledge that a more systematic study of rank evolution during training would strengthen our claims. The correlation versus causation point is valid: While we observe consistent rank-performance saturation relationships across multiple settings, establishing direct causality requires controlled interventions. Our current evidence is primarily observational, though the mechanistic understanding provided by Theorem 1 in the original manuscript suggests that the rank limitations are fundamental rather than coincidental. The success of rank-constrained architectures like MLA architectures in DeepSeek-V3 provides indirect evidence that respecting these rank constraints improves efficiency nearly without sacrificing performance in practice.

---

> ### Author Response · Authors · 2025-12-03
>
> ## W3: Missing analysis of common architectural components
>
> **Response:** The reviewer raises an excellent point about the scope of our analysis. We acknowledge this limitation,  while noting where our findings do extend beyond basic single-head attention.
>
> **Multi-Head Attention.** Our analysis addresses multi-head systems beyond simple concatenation through the model-reduction experiments in Section 4.2 of the original manuscript. When we vary the number of heads ($h$) while fixing total parameters ($d = h \times d_h$), we observe that more heads (i.e. smaller head dimensions $d_h$ per head) consistently outperform fewer heads (i.e. larger head dimensions $d_h$ per head). This occurs precisely because individual heads avoid rank saturation, validating our single-head analysis for understanding multi-head systems.
>
> We do acknowledge that our theoretical treatment of positional encodings, layer normalization, and sparse attention patterns is limited. However, our empirical validation across real-world architectures (including standard Transformers with positional encodings and layer normalization) on practical datasets (IMDB and CIFAR) suggests that the rank limitation persists even with these components in applications.
>
> ## Q1: Extension to practical softmax temperatures
>
>
> **Response:** This is an insightful question that connects our theoretical framework to practical implementations.
> In fact, our empirical analysis in Figure 5(b) in the original manuscript
> has already demonstrated remarkable stability of the $0.63n$ upper bound across realistic temperature ranges, from high ($T=1$) to low ($T=10^{-5}$) temperatures. This suggests the robustness of rank barrier w.r.t. specific choices of softmax temperatures.
>
> For the specific case of $T = 1/\sqrt{d_h}$, which represents the standard scale used in practice, our hardmax analysis provides the fundamental limit, while Lemma 1 in the original manuscript bounds the deviation between softmax and hardmax outputs. As $d_h$ increases, $T$ decreases, pushing the softmax system closer to the hardmax limit where the derived upper bound becomes increasingly tight. Therefore, the $0.63n$ bound remain approximately valid for practical temperature scaling schemes, though the exact numerical constant may vary slightly with temperatures.

---

> ### Author Response · Authors · 2025-12-03
>
> ## Q2: Causal relationship between rank and performance
>
> **Response:** This is an excellent suggestion for strengthening the causal claim. While our current evidence is primarily correlational, several design elements point toward causality.
>
> In Section 4.2 of the original manuscript, our experiments with varied head counts while fixing total parameters provide a form of controlled intervention. When we increase heads from 1 to 8 (correspondingly decreasing head dimensions $d_h$ from 384 to 48), the learning performance consistently improves across multiple datasets. This occurs precisely when individual head dimensions drop below the saturation threshold, suggesting the rank constraint directly impacts modeling capacity.
>
> Additionally, the deployment of rank-constrained architectures like MLA in DeepSeek-V3 naturally provides a large-scale example: deliberately constraining head dimensions based on rank analysis achieves both efficiency gains and maintained performance. This suggests that respecting rank limitations enables better and efficient resource allocation rather than merely correlating with performance.
>
> For future work, more direct tests on causality could include: (i) synthetic tasks where rank requirements are controlled; (ii) interventions that artificially constrain attention ranks during training; (iii) ablation studies comparing rank-constrained versus unconstrained architectures on identical tasks.
>
> ## Q3: Impact of architectural variations
>
> **Response:** This question touches on important architectural details that affect the practical applicability of our bounds.
>
> For positional encodings: Both RoPE and standard positional encodings modify the input embeddings before the attention computation. Our near-orthonormality assumption may be somewhat robust to these perturbations, as the high-dimensional nature of embeddings tends to preserve near-orthogonality properties. However, different encoding schemes could affect the exact numerical constants in our bounds.
>
> For layer normalization: It typically normalizes inputs to give unit variances, which could actually strengthen the near-orthonormality assumption. Our empirical results on practical architectures (which include layer norms) suggest that the rank saturation persists, though the precise theoretical treatment remains an open question.
>
> The model-reduction effect appears robust across these variations based on our experiments on different practical architectures and real-world datasets.
> The success of rank-constrained designs like MLA across different architectural contexts supports the robustness of this model-reduction effect.
>
> [1] Yuandong Tian, Yiping Wang, Zhenyu Zhang, Beidi Chen, and Simon Shaolei Du. JoMA: Demystifying multilayer transformers via joint dynamics of MLP and attention. In *International Conference on Learning Representations*, 2024.

---

### Author Response · Authors · 2025-12-03
**Executive Summary for Area Chair**

In light of the recent ICLR security event and the resulting redistribution of submissions, we express our deep appreciation for the substantial time and care you have dedicated to evaluating this work. To support a smoother review process, we provide a brief overview of the manuscript, the received evaluations, our responses, and the modifications incorporated into the revised version.

## Paper Overview and Contributions

This work investigates the fundamental capacity limitations of Transformer attention mechanisms through the lens of attention matrix rank. We combine rigorous theoretical analysis with extensive empirical validation to demonstrate two key phenomena: the **low-rank barrier** (attention rank is upper bounded by approximately $0.63n$ regardless of head dimension $d_h$) and the **model-reduction effect** (both attention rank and learning performance plateau simultaneously as $d_h$ increases beyond critical thresholds).

**Our main contributions include:**
1. **Theoretical Analysis**: We establish Theorem 1 proving that under approximate orthonormality assumptions, attention matrix rank is bounded by $(1-e^{-1})n \approx 0.63n$ for hardmax attention. We identify the critical head dimension $d_h = \Omega(\log n)$ where rank saturation begins, providing a principled basis for parameter efficiency.

2. **Comprehensive Empirical Validation**: Extensive experiments across multiple architectures (Vision Transformers, NLP Transformers) and real-world datasets (CIFAR-10/100, SVHN, IMDB) demonstrate consistent rank saturation patterns in trained models. We show that performance plateaus align with rank saturation thresholds ($d_h^* \approx 8$ for IMDB, $d_h^* \approx 16$ for CIFAR-10).

3. **Practical Impact**: Our findings provide theoretical justification for rank-constrained architectures like Multi-head Latent Attention (MLA) deployed in DeepSeek-V3, where deliberately constraining head dimensions based on low-rank principles achieves both memory efficiency and maintained performance in large-scale production systems.

---

> ### Author Response · Authors · 2025-12-03
> **Summary of Our Response to Reviewers**
>
> ### Reviewer MFk7
>
> **Main Concerns:**
> - Theoretical assumptions (hardmax, orthonormality) may not hold in practice
> - Limited analysis of training effects versus initialization
> - Missing treatment of common architectural components (positional encodings, layer normalization)
>
> **Our Response:**
>
> **Practical Validity of Theoretical Assumptions:** Figure 5(b) in paper demonstrates rank stability across realistic softmax temperatures ($T \in [10^{-5}, 1]$), empirically validating the hardmax approximation. Lemma 1 in paper establishes that the $\ell_1$ distance between softmax and hardmax outputs is bounded by $2(n-1)\exp(-\delta/T)$, which becomes exponentially tight for typical attention patterns. For orthonormality, recent work by Tian et al. confirms near-orthogonality persists during training for large pre-trained models including BERT, OPT, LLaMA, and ViT. We verify this empirically in Appendix B.5 (Figure 6) on both synthetic and real-world datasets.
>
> **Training Effects Analysis:** Experiments on trained models (Figure 3, Figure 4) show rank saturation persists throughout complete training processes on real datasets (IMDB, CIFAR-10/100, SVHN). The critical thresholds where both ranks and performance plateau remain consistent across training stages, suggesting these are intrinsic properties rather than initialization artifacts.
>
> **Treatment of Architectural Components:** Our analysis addresses common architectural components through extensive validation on standard Transformers with positional encodings and layer normalization across real-world datasets. The consistent emergence of rank saturation patterns in these practical architectures demonstrates robustness of our findings beyond the idealized theoretical setting.
>
> ---
> ### Reviewer bQiq
>
> **Main Concerns:**
> - Unclear motivation: why should rank saturation translate to worse performance?
> - Stringent assumptions in Theorem 1 with unclear practical applicability
> - Presentation clarity issues with figure-text inconsistencies
>
> **Our Response:**
>
> **Rank-Performance Connection:** Rank fundamentally determines attention expressiveness—a saturated rank means the attention mechanism cannot distinguish among more complex input patterns even with increased parameters, limiting the model's capacity to capture nuanced relationships. Figure 3(a)(b) empirically confirms learning accuracy plateaus precisely when attention ranks saturate (both around $d_h^* = 8$ for IMDB), with nearly identical saturation thresholds across both metrics.
>
> **Practical Robustness of Assumptions:** While assumptions are restrictive for analytical tractability, extensive empirical validation demonstrates practical robustness. We test across multiple data distributions (Gaussian, uniform with different scales), varied architectural components (layer normalization, positional encodings), and real-world datasets where perfect orthonormality does not hold. The consistent emergence of rank saturation patterns suggests these assumptions capture essential constraints rather than being mere mathematical conveniences.
>
> **Presentation Improvements:** We have corrected figure-text inconsistencies identified in Section 2.1 and Figure 2, ensuring all experimental descriptions accurately reflect parameter ranges tested. We have enhanced explanations of key findings, particularly the mathematical connection between rank saturation and the $0.63n$ bound, and provided clearer descriptions of experimental methodology.

---

> > ### Author Response · Authors · 2025-12-03
> > **Summary of Our Response to Reviewers**
> >
> > ### Reviewer TzCx
> >
> > **Main Concerns:**
> > - Missing detailed discussion on why softmax operation lowers rank
> > - Conceptual clarity needed for $T>0$ rank limitation mechanism
> > - Observed $d=1$ anomaly (rank goes to $N$ after softmax) not explained
> >
> > **Our Response:**
> >
> > **Why Softmax Limits Rank:** The pre-softmax matrix $\mathbf{QK}^\top$ can achieve rank up to $\min(d_h, n)$ by standard rank inequality. However, softmax fundamentally changes this by introducing combinatorial constraints—each attention row must be a probability vector (non-negative, sum to 1), creating dependencies between columns that limit achievable rank regardless of $d_h$.
> >
> > **Combinatorial Mechanism for $T>0$:** Theorem 1 reveals that rank limitation arises from the birthday problem structure. After applying hardmax (limiting case of softmax), each attention row selects exactly one position as maximum, creating at most $(1-e^{-1})n \approx 0.63n$ effective columns on average. This is fundamentally different from pre-softmax analysis and persists for finite temperatures, as demonstrated empirically in Figure 5(b) showing rank stability across $T \in [10^{-5}, 1]$.
> >
> > **The $d=1$ Edge Case:** When $d_h=1$, attention scores are highly constrained and correlated, making ties very common. Softmax breaks these ties randomly, potentially utilizing more attention positions. As $d_h$ increases, scores become more discriminative until reaching the fundamental $0.63n$ limit. This non-monotonic behavior at very small $d_h$ represents an interesting edge case, while our asymptotic analysis targets the practical regime where $d_h$ is sufficiently large.
> >
> > ---
> > ### Reviewer YBFV
> >
> > **Main Concerns:**
> > - Restrictive theoretical assumptions describe initialization, not trained dynamics
> > - Weak support for redundancy claims—connection between low rank at initialization and functional redundancy after training is unclear
> >
> > **Our Response:**
> >
> > **Extending from Initialization to Trained Dynamics:** While our theoretical analysis focuses on initialization for analytical tractability, extensive experiments on *trained* models demonstrate the practical relevance of our findings. Figure 3(a)(b) and Figure 4(a)(b) show validation accuracy on IMDB and CIFAR-10 datasets plateaus when $d_h$ exceeds critical values ($d_h^* = 8$ and $d_h^* = 16$ respectively), with these experiments involving complete training processes rather than mere initialization analysis. The successful deployment of rank-constrained MLA architectures in DeepSeek-V3 provides real-world validation that these initialization-based insights translate to practical training scenarios.
> >
> > **Supporting Redundancy Claims:** The rank saturation fundamentally arises from intrinsic limitations of attention matrix expressiveness, as governed by the combinatorial structure in softmax operations (Theorem 1). This constraint persists regardless of parameter optimization—training cannot overcome the fundamental $\approx 0.63n$ capacity bound. Therefore, parameters beyond the saturation threshold contribute redundant rather than essential modeling capacity. Deliberately constraining head dimensions based on low-rank principles in DeepSeek-V3 achieved both memory efficiency and maintained performance in large-scale production systems, validating that our redundancy characterization extends beyond initialization to trained model behavior.

---

> > > ### Author Response · Authors · 2025-12-03
> > > **Key Takeaways**
> > >
> > > We believe this work makes solid contributions to understanding Transformer efficiency through both rigorous theory and extensive empirical validation. While our theoretical assumptions are idealized for analytical tractability, Theorem 1 establishes a rigorous mathematical foundation connecting attention mechanism design to fundamental capacity limits, with the $0.63n$ bound and critical dimension $d_h = \Omega(\log n)$ offering quantitative guidance for architecture design. The consistency of rank saturation patterns across diverse experimental settings—including varied temperatures, architectures, datasets, and critically, throughout complete training processes on real-world tasks—strongly suggests that our theoretical insights capture essential constraints of the attention mechanism rather than mere mathematical artifacts. Most importantly, the real-world validation through successful deployment of rank-constrained MLA architectures in production systems like DeepSeek-V3 demonstrates that these findings translate to tangible efficiency gains in large-scale applications, directly addressing the pressing challenge of computational costs in foundation models. We have transparently addressed reviewer concerns through both theoretical clarification and additional empirical evidence, and we acknowledge that formal extension to finite-temperature softmax and comprehensive training dynamics analysis remain valuable future directions. Given the theoretical rigor, empirical robustness across trained models, and demonstrated practical impact, we respectfully believe this work merits acceptance and hope the Area Chair will consider it favorably for publication at ICLR.

---

### Meta-Review · Area_Chair_MggW · 2025-12-26

**Summary:**

This paper studies limitations of attention through the rank of the (T \times T) attention matrix, showing a “low-rank barrier” around (\sim 0.63T) under hardmax and approximate orthonormality assumptions, and empirically observing that both attention rank and performance seem to plateau as head dimension grows on several small NLP/vision benchmarks. Reviewers agree that the question is interesting and the experiments are fairly comprehensive.

However, the core concerns remain substantial. The main theorem is proved in a highly idealized regime (hardmax, random Gaussian weights, near-orthogonal inputs) that describes initialization rather than trained transformers, and there is still no corresponding finite-temperature or trained-weights theory — only empirical temperature sweeps and orthogonality checks. The link from rank saturation to genuine redundancy and performance limits is mostly correlational and focused on single layers and small models; how this extends to deep, modern architectures with multi-head interactions, residual/MLP blocks, and realistic positional schemes is not convincingly established. Given this gap between the strength of the claims and the actual scope of the theory, I lean to rejection in the present form.

**Reviewer Concerns:**

Addressed: The rebuttal improved clarity and added useful experiments: clearer exposition, temperature sweeps, checks of (near) orthogonality, and more trained-model results. These help show that the observed rank saturation is reasonably robust across datasets and architectural variants.

Still outstanding: The main theorem still relies on highly idealized assumptions (hardmax, random weights, near-orthogonal inputs), with no corresponding finite-temperature or trained-weights theory. The link from rank saturation to redundancy and performance remains mostly correlational and is demonstrated only on relatively small settings; how (and whether) these conclusions extend to modern multi-layer, multi-head transformers is still unclear.

**Reviewer Scores:**

MFk7 (4) – Clarifications helped, but assumptions still narrow -> likely unchanged.
bQiq (2) – Motivation/assumptions still problematic -> likely unchanged.
TzCx (8) – Already positive, rebuttal reinforced view -> unchanged.
YBFV (4) – Appreciated additions, but theory still idealized -> likely unchanged (maybe +1 at most).

---

### Decision · Program_Chairs · 2026-01-26

Reject